# Aerosol variability over oceans using micro-pulse lidar and photometer: Insights from TRANSAMA ship-based campaign

Maria Fernanda Sanchez-Barrero[1], Philippe Goloub[1], Luc Blarel[1], Ioana Elisabeta Popovici[1,2],

Benjamin Torres[1], Gaël Dubois[1], Thierry Podvin[1], Fabrice Ducos[1], Romain de Filippi[1], Michaël

Sicard[3,4], Viviane Bout Roumazeilles[5], Charlotte Skonieczny[6]

[1]Univ. Lille, CNRS, UMR 8518 - LOA - Laboratoire d'Optique Atmosphérique, F-59000 Lille, France

[2]R&D Department, CIMEL Electronique, 75011 Paris, France

[3]Univ. La Réunion, CNRS, UMR 8105 - LACy - Laboratoire de l'Atmosphère et des Cyclones, Météo-France, 97400 Saint-Denis de La Réunion, France

[4]CommSensLab-UPC, Universitat Politècnica de Catalunya, Barcelona, 08034, Spain

[5]Univ. Lille, CNRS-UMR8217, Laboratoire Géosystèmes, 59655 Lille, France

[6]Univ. Paris-Saclay, CNRS, UMR 8148 - GEOPS - Laboratoire Géosciences Paris-Saclay, 91405 Orsay Cedex, France

*Correspondence to*: Maria F. Sanchez-Barrero (mariafernanda.sanchezbarrero@univ-lille.fr)

**Abstract.** The TRANSAMA campaign (Transit to AMARYLLIS-AMAGAS oceanographic cruise), conducted aboard the research vessel *Marion Dufresne II* assessed instrument performance and investigated aerosol properties during its transit from La Reunion Island to Barbados (April–May 2023). A set of remote sensing instruments, including two CE318-T Sun-sky-lunar photometers and a CE370 single-wavelength elastic lidar, was deployed under the MAP-IO (Marion Dufresne Atmospheric Program–Indian Ocean) framework. Performance assessments of the deployed instrumentation support the development of coupled lidar–photometer systems for shipborne atmospheric observations, while acknowledging current detection limits. Synergistic observations provided vertically resolved aerosol properties, such as extinction coefficients, alongside atmospheric structure, highlighting the marine boundary layer (MBL) top at $800 \pm 300$ m. While the photometer observations revealed clean atmospheric conditions over the South Atlantic ($AOD(440) = 0.08 \pm 0.04$), thin aerosol layers above the MBL were identified as long-range transported residual biomass-burning-urban aerosols from Southern Africa with effective LR of $33 \pm 12$ sr. Cloud layers covering a large range of altitudes (up to 16 km) were observed in 53% of the lidar profiles, with a higher frequency at lower altitudes, where aerosol layers were more frequently detected. These findings emphasize the impact of continental aerosols on remote oceanic regions and demonstrate the capabilities of synergistic lidar–photometer measurements for advancing our understanding of aerosol variability, cloud formation, and climate processes over the oceans.

# 1 Introduction

Aerosols play a critical role in atmospheric processes, influencing climate, air quality and human health. Despite substantial
advancements in aerosol research, significant uncertainties remain associated to aerosol radiative impacts (Boucher et al.,
2013; Szopa et al., 2021). Marine aerosols, primarily sea spray consisting of sea salt and organic matter, dominate aerosol
content over the oceans, which cover more than 70% of the Earth's surface. Wind speed is the primary factor driving the
generation of natural marine aerosols (Flamant et al., 1998; Smirnov et al., 2012; Sun et al., 2024). These aerosols mainly
exhibit high scattering efficiency and hygroscopicity enforcing cloud formation (Pierce and Adams, 2006; Jaeglé et al., 2011;
Burton et al., 2013; Bohlmann et al., 2018). Their interaction with continental aerosols, such as Saharan dust transported over
the Atlantic (Kanitz et al., 2014; Bohlmann et al., 2018; Barreto et al., 2022), or biomass burning aerosols in the Southern
hemisphere (Duflot et al., 2011; Chazette et al., 2019; Formenti et al., 2019; Ranaivombola et al., 2025), are critical for
understanding climate feedbacks.

Marine optical properties have been studied using passive remote sensing instruments deployed aboard satellites and ground-
based platforms located in remote islands and coastal regions or during ship-based campaigns. Observations consistently show
that aerosol optical depth (AOD) over pristine oceanic regions, such as the South Atlantic, South Indian, and Pacific Oceans,
is typically lower than 0.1, providing insights into pre-industrial conditions and aerosol-climate interactions (Hamilton et al.,
2014; Koren et al., 2014; Mallet et al., 2018; Duflot et al., 2022). Even small changes in such low aerosol loadings can
significantly impact cloud formation, as suggested by Koren et al. (2014). However, aerosol products derived from satellite
data are subject to non-negligible uncertainties, with AOD retrieval errors exceeding 0.04 at 550 nm, whereas ground-based
measurements exhibit considerably lower uncertainties (0.01–0.02 across different wavelengths). Validation of satellite-
derived aerosol products through ground-based observations over open ocean regions is essential (Gupta et al., 2016; Chen et
al., 2020). Yet, discrepancies between satellite-derived AOD and ground-based measurements highlight the lack of aerosol
observations over oceans (Mallet et al., 2018; Ranaivombola et al., 2023 and references therein).

Ship-based observatories offer a unique opportunity to fill these observational gaps. Sun photometer maritime observations
were deployed by the Maritime Aerosol Network (MAN; Smirnov et al., 2009), an initiative of the world-wide open-access
photometer network AERONET (AErosol Robotic NETwork; Holben et al., 1998). MAN collects AOD at five wavelengths
manually with Microtops II handheld Sun photometers aboard ship cruises (Duflot et al., 2011; Smirnov et al., 2012). While
highly valuable, these measurements are limited to the availability of observers, preventing unattended and continuous
operation.

In recent years, the multispectral Sun-sky-lunar CIMEL CE318-T photometer (Barreto et al., 2016), widely used and designed
by the French company Cimel, has been fully adapted for automatic observations onboard ships (Yin et al., 2019). This
adaptation has been tested and refined through several maritime campaigns (Torres et al., 2025). In 2021, the first operational

ship-adapted CE318-T photometer for continuous aerosol monitoring was installed aboard the Marion Dufresne II (Torres et al., 2025), the largest oceanographic research vessel (RV) of the French fleet operated by the IFREMER (Institut Français de Recherche pour l'Exploitation de la Mer, https://www.ifremer.fr/, last access: December 6 2024). Additional historical and technical developments are detailed in Torres et al. (2025). Similarly, the PLASMA (Photomètre Léger Aéroporté pour la Surveillance des Masses d´Air; Karol et al., 2013) sun-photometer, developed and deployed by the Laboratoire d'Optique Atmospherique (LOA, Lille University) has been deployed aboard aircraft and vehicles for high-temporal-resolution measurements (Mascaut et al., 2022; Popovici et al., 2018, 2022; Sanchez Barrero et al., 2024). PLASMA provides AOD data every 10 s (vs. 3 minutes of CE318-T photometer) but does not perform sky measurements, and it is not yet adapted for continuous operation under rough sea conditions. Both instruments follow AERONET processing protocols.

The MAP-IO (Marion Dufresne Atmospheric Program–Indian Ocean) platform (http://www.mapio.re/, last access: December 6, 2024) is a French National Instrumental facility aboard the *Marion Dufresne II* RV and it is part of ACTRIS-FR (Aerosol, Clouds and Trace gases Research Infrastructure-France; https://www.actris.fr/, last access: October 25, 2025). The observatory integrates a radiometer (measuring solar flux), in-situ instruments for gas and aerosol monitoring and the ship-adapted CE318-T photometer. Since 2021, this observatory has supported ocean-atmosphere studies (Tulet et al., 2024), and its success has led to the installation of ship-adapted CE318-T photometers on additional research vessels (Torres et al., 2025).

Lidar systems aboard mobile vectors (car, aircraft and ship) have demonstrated their ability to complement satellite observations (Burton et al., 2013; Warneke et al., 2023), assess air quality in urban-rural transitions and complex topographies (Royer et al., 2011; Chazette et al., 2012; Dieudonné et al., 2015; Shang et al., 2018; Popovici et al., 2018, 2022; Chazette and Totems, 2023; Sanchez Barrero et al., 2024), and characterize transported aerosols such as dust or smoke over oceans (Duflot et al., 2011; Burton et al., 2013; Kanitz et al., 2014; Yin et al., 2019) and land (Popovici et al., 2018, 2022; Chazette and Totems, 2023; Warneke et al., 2023; Sanchez Barrero et al., 2024). Despite the advances in lidar conception, deploying high-power systems like multi-wavelength Raman or High Spectral Resolution Lidars (HSRL) for unattended continuous in-movement operations remains challenging due to size, environmental sensitivity and mechanical stability, restricting measurements to field campaigns (Burton et al., 2013; Müller et al., 2014; Bohlmann et al., 2018; Yin et al., 2019). Compact, automatic micropulse lidar systems offer a practical alternative, especially when coupled with column-integrated photometer observations to constrain aerosol properties (Chazette, 2003; Mortier et al., 2016; Popovici et al., 2018, 2022; Sanchez Barrero et al., 2024). Popovici et al. (2018) first demonstrated a mobile integrated system, combining a CE370 single-wavelength elastic lidar and the PLASMA photometer, capable of on-road aerosol measurements, demonstrating its versatility for aerosol characterization during campaigns (Popovici et al., 2018, 2022; Warneke et al., 2023; Sanchez Barrero et al., 2024).

The oceanographic cruise AMARYLLIS-AMAGAS, conducted aboard the *Marion Dufresne II* RV, took place from 16 May to 3 July 2023 (Govin et al., 2024). The Franco-Brazilian campaign essentially aimed to study the Amazon's role in past climates using core sediments. TRANSAMA (Transit to AMARYLLIS-AMAGAS) campaign emerged as initiative to valorize

the transit from Le Port, La Reunion Island (*Marion Dufresne II* main port; 20.94 °S, 55.29 °E) to Bridgetown, Barbados (mission's starting port, 13.10 °N, 59.63 °W). TRANSAMA employed a combination of remote sensing instruments, including the installation of a second CE318-T photometer and the CE370 elastic lidar, to perform operational assessments on the instrumentation embarked and study the aerosol properties along the ship's route covering the South Indian and the Atlantic oceans.

The primary objective of this study is to demonstrate the synergistic application of micropulse lidar and ship-adapted photometers assessing the performance and limitations of continuous, autonomous observations through TRANSAMA campaign (April–May 2023). Additionally, the study characterizes the vertical distribution of aerosols and clouds across the South Indian and Atlantic Oceans, with emphasis on long-range transported continental aerosol plumes affecting remote marine regions. By bridging observational gaps over oceans, this approach contributes to ongoing discussions on aerosol-climate interactions, satellite calibration and validation, and emphasizes the need for comprehensive observational strategies in marine environments.

This work is structured as follows: Section 2 describes the mobile remote sensing instruments used in this study and their setup aboard the Marion Dufresne RV, with Section 2.1 detailing the CE370 lidar, Section 2.2 the CE318-T photometer, and Section 2.3 ancillary measurements. Section 3 presents the methods for data processing for both lidar (Section 3.1) and photometer (Section 3.2) and their combination to derive aerosol properties (Section 3.3). Section 4 presents results and discussion of the environmental conditions (Section 4.1), instrumental assessments (Section 4.2) and observed aerosol properties during the campaign (Section 4.3).

## 2 Remote sensing instrumentation

Measurements have been performed using both passive (photometer) and active (lidar) remote sensing instrumentation (Figure 1). Raw data were transmitted to the LOA server (SNO/PHOTONS, University of Lille), the photometers every 15 minutes and lidar every hour, for data-processing and visualization.

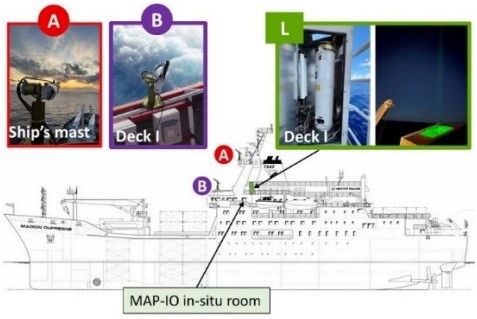

**Figure 1.** Distribution of the remote sensing instrumentation aboard *Marion Dufresne* RV during the TRANSAMA campaign. The location of the two CE318-T photometers (A and B) and the CE370 single-wavelength lidar (L) are indicated.

## 2.1 Single-wavelength lidar

The **CE370 lidar** is an eye-safe, single-wavelength micro-pulse elastic lidar operating at 532 $nm$ with a 20 $\mu J$ pulse energy at a 4.7 $kHz$ repetition rate. Designed by Cimel Electronique, it utilizes a shared transmitter/receiver telescope (diameter of 20 $cm$ and half field of view of 55 $\mu rad$) connected via an 10 $m$ optical fiber to its laser source and control system (laser divergence of 240 $mrad$). Photon-counting with an avalanche photodiode (APD) detects the elastic backscattered signals, enabling monitoring of aerosol and cloud properties in the troposphere with a vertical resolution of 15 m and one minute accumulation time (Pelon et al., 2008; Mortier et al., 2013; Popovici et al., 2018).

For field campaigns, the CE370 lidar coupled with a photometer has been deployed on ground-based mobile platforms. The system has demonstrated its capability to characterize vertical aerosol properties under diverse conditions, including highly polluted urban areas, proximity to fire sources, and rough terrain during platform movement (Popovici et al., 2018, 2022; Hu et al., 2019; Sanchez Barrero et al., 2024).

The laser source was housed in the temperature-controlled MAP-IO in-situ room, featuring vibration isolation and movement dissipation structure. The transmission/reception telescope was installed on Deck I within an enclosure equipped with an automatic blind system to prevent direct sunlight exposure and regulated temperature to avoid condensation on the window (Fig. 1). However, the constant sea spray produced by the waves hitting the vessel deposited and dried on top of the lidar window resulting in the attenuation of the laser beam. A frequent cleaning was reinforced to avoid the laser attenuation (More details are discussed in Sect. 4.2.1).

## 2.2. Photometers

The CIMEL **CE318-T Sun-sky-lunar photometer** (Barreto et al., 2016; Giles et al., 2019) performs automated direct sun and lunar measurements across nine spectral channels (340–1640 nm). It measures aerosol optical depth (AOD, accuracy ± 0.01) and extinction Angstrom exponent (EAE), while multi-angular sky radiance measurements enable retrievals of aerosol microphysical properties, such as volume size distribution, refractive index, and single-scattering albedo (Dubovik and King, 2000; Sinyuk et al., 2020). For ship-borne applications, the CE318-T is equipped with GPS and compass modules to correct for ship's attitude (heading, pitch and roll), ensuring accurate Sun and Moon tracking (Yin et al., 2019). Since January 2021, it has been operational aboard the *Marion Dufresne II* RV as part of the MAP-IO project (Tulet et al., 2024) from which successful three years of continuous measurements were presented by Torres et al. (2025).

The CE318-T ship-borne photometer (# 1273) operational since 2021 aboard the *Marion Dufresne II*, photometer *A* hereafter, was installed on the ship's main mast (Fig. 1) to minimize potential obstacles on tracking. Its continuous observations revealed technical challenges, such as wrong Sun-Moon tracking due to strong waves combining with a rather slow communication between the control/acquisition software and GPS-compass module. To address these challenges, a second CE318-T

photometer installed during TRANSAMA was used for test and validation (see Section 4.2). The second photometer (# 1243), photometer *B* hereafter, was installed on deck I (Fig. 1) and accounted for an updated software to improve the tracking response and slightly modified design (shorter collimator to enlarge the field of view). In addition, the prototype of PLASMA-3 photometer was installed on deck I, next to photometer B, for short test measurements of the Moon tracking, the instrument operated only intermittently, and the results are not included in this work.

## 2.3 Ancillary measurements

In the framework of MAP-IO project, ancillary data is provided from a Vaisala weather transmitter (WXT530) installed in the main mast. It provides meteorological measurements every 5 seconds, meaning a spatial resolution of 25-75 m considering vessel speed of 5-15 m/s (18-54 km/h). Measurements include temperature (accuracy: $\pm 0.3\ °C$), pressure (accuracy: $\pm 0.5\ hPa$ at 0-30 $°C$), relative humidity (accuracy: $\pm 3$ %), wind speed (rel. accuracy: $\pm 3$ % $at\ 10\ m/s$) and wind direction (accuracy: $\pm 3\ °\ at\ 10\ m/s$). The wind measurements reflect the apparent wind speed and direction. To obtain the true wind speed and direction, vectorial addition is employed, using the vessel's heading and speed to accurately account for the motion of the ship.

## 3 Methodology

This section focuses on the methodology employed to study aerosol properties and the atmospheric structures. The lidar data processing is presented in Section 3.1, including the description of detection limits (Sect. 3.1.1) and detection of marine boundary layer (MBL) top, aerosol and cloud layers (Sect. 3.1.2). The photometer data processing is briefly presented in Section 3.2. The inversion method used to derive aerosol properties combining lidar and photometer are presented in Section 3.3, including a description of uncertainties calculation.

## 3.1 Lidar data processing

The detected backscattered lidar signal is represented by equation (1) (Kovalev and Eichinger, 2004), with the wavelength dependency omitted for simplicity:

$$RCS(r) = C_L[\beta_{mol}(r) + \beta_{aer}(r)]exp\left(-2\int_0^r \alpha_{mol}(r')dr'\right)exp\left(-2\int_0^r \alpha_{aer}(r')dr'\right) \tag{1}$$

$$\beta_{att}(r) = \frac{RCS(r)}{C_L} \tag{2}$$

The right side of Eq. (1) describes the range-corrected signal (RCS, in photons $s^{-1}$ m²) in terms of the atmospheric optical properties, calibrated using the constant $C_L$ (in photons $s^{-1}$ m³ sr). The backscatter coefficient $\beta(r)$ (in $m^{-1}sr^{-1}$) and the extinction coefficient $\alpha(r)$ (in $m^{-1}$) are distinguished by subscripts for molecular (mol) and aerosol (aer) contributions. Thus, the total attenuated backscatter $\beta_{att}(r)$ (in $m^{-1}sr^{-1}$) defined in Eq. (2) accounts for both contributions.

Key instrumental corrections—including background subtraction (B), after-pulse and dead time effects, overlap function (O), and range dependence ($r^2$)—are applied to derive the RCS, following established methods (Pelon et al., 2008; Mortier et al., 2013; Popovici et al., 2018).

### 3.1.1 Data quality control and normalization

Quality control of lidar data involves the Rayleigh fit procedure (Freudenthaler et al., 2018), that is, normalizing RCS(r) to a modeled molecular profile $\beta_{mol}(r)\exp\left(-2\int_0^r \alpha_{mol}(r')\,dr'\right)$ in a reference zone ($r_{ref}$) assumed to be free of aerosols ($\beta_{aer}(r_{ref}) = 0$). Molecular backscatter and extinction coefficients are calculated using temperature and pressure profiles from standard models or radiosonde data. The slope index, defined as the ratio of normalized RCS to molecular profile slopes in the reference zone, evaluates lidar performance (a slope index closes to 1 indicates good performance). When photometer AOD data is available and clean conditions are met, the lidar calibration is determined ($C_L$) (Sanchez-Barrero et al., 2024 and references therein). However, during the campaign frequent laser attenuation due to sea spray deposition on the lidar window prevented a consistent lidar constant for the entire dataset (more details are presented in Sect. 4.2.1). Therefore, a normalized relative backscatter (NRB) was defined instead (Lopatin et al., 2013).

$$NRB(r) = \frac{RCS(r)}{\int_{r_o}^{r_{max}} RCS(r')dr'} \tag{3}$$

Where $r_o$ is the lower detection limit (Sect. 3.1.2), and $r_{max}$ is set at 10 km consistent with prior studies (Welton and Campbell, 2002; Lopatin et al., 2013, 2024). In particular the NRB profiles improve studies on atmospheric structure and data visualization, but do not have an impact on the later inversion procedure (Section 3.3) if the detection limits are not strongly affected.

### 3.1.2 Detection limits

The detection limits of the CE370 lidar primarily depend on signal-to-noise ratio (SNR) and the overlap function as discussed in previous studies (Popovici et al., 2018; Sanchez Barrero et al., 2024). The lower detection limit (blind zone) was defined at 350 m, where overlap function uncertainties exceed 20 %. The upper detection limit, determined by SNR < 1.5, varies between 10 and 18 km due to background noise from solar irradiance and the laser beam attenuation.

Significant motion of the RV caused by rough seas, resulting in high variation in pitch or roll, introduces negligible uncertainties in the lidar profiling. For example, a 5-degree tilt in any direction causes altitude errors of 0.3% and horizontal shifts of 0.1% relative to the zenith. Furthermore, the system's combination of a wide laser divergence and a narrow telescope field of view improve detection at lower altitudes and combined with a robust opto-mechanical design minimize misalignments due to platform motion. Additionally, the lidar's slightly offset viewing angle reduces direct reflections from clouds, enhancing measurement quality.

Once the lidar detection limits have been defined and the NRB profiles obtained, the following subsection outlines the methodology implemented to determine the atmospheric structure from each lidar profile measured.

### 3.1.3 Atmospheric structure detection

Atmospheric structure detection is applied individually to each one-minute lidar profile to capture transient cloud events. The methodology follows BASIC algorithm (Mortier, 2013), developed by the LOA, and STRAT (Morille et al., 2007), a dedicated software for atmospheric structure detection. Molecular regions are identified following the STRAT guidelines, while aerosol and cloud layers are detected via the gradient method described in BASIC. The layer base and peak are determined by the first derivative of the NRB signal with threshold to exclude wrong detection. An initial filtering using the threshold (T1) is defined

as $NRB(r_{peak}) - NRB(r_{base}) > T1 \times N(r_{base})$, where N is the noise level. The distinction between aerosol and cloud layers is determined using a second threshold (T2), where the ratio $NRB(r_{peak})/NRB(r_{base}) > T2$ indicates the presence of a cloud layer.

The MBL top is detected using the Haar wavelet covariance transform (WCT) method (Flamant et al., 1997; Brooks, 2003; Baars et al., 2008; Granados-Muñoz et al., 2012). The maximum in the WCT profile in the range 0.35-3 km identifies the MBL

top. Clouds often form at the top of the boundary layer (Stull, 1988; Bechtold and Siebesma, 1998; Kotthaus et al., 2023); thus, in cases where low clouds are detected, the MBL detection is constrained to the cloud base.

For this study, the threshold T1 was set to 5 and T2 was set to 4, based on guidance from previous works (Morille et al., 2007; Mortier, 2013; Mortier et al., 2016). If the upper detection limit of the profile, falls within the blind zone (below 350 m), the profile is considered saturated, which could result from condensation, fog or very low-level clouds.

Having defined the methods used for lidar data processing, the following section briefly describes the photometer data processing.

### 3.2 Photometer data processing

Collected data from the ship-adapted CE318-T photometer are transmitted to PHOTONS (University of Lille) and subsequently to the NASA AERONET server, where they are processed. AERONET's data processing protocols enable near

real-time (NRT) aerosol monitoring with rigorous quality screening (Smirnov et al., 2000; Holben et al., 2006; Giles et al., 2019). These protocols, identical to those applied at land-based AERONET sites, are implemented using the Version 3 algorithms described by Giles et al. (2019).

The instrument performs Sun-Moon triplet measurements, meaning that a sequence of observations at all wavelengths is taken three times at 30-second intervals, resulting in a one-minute averaged measurement (data Level 0 or L0). Most L0 data quality

checks operate on voltage signal, expressed as the integer digital counts or digital number (DN). Data are mainly promoted to

Level 1 (L1) after verification of the correct Sun/Moon tracking, based on the triplet variability criteria. This means that the root-mean-square error (RMSE) of the DN triplet measurements relative to one-minute averaged must remain below a threshold, defined at 16 %. Additional electronic tests are also considered to prevent erroneous measurements from reaching L1 (Giles et al., 2019).

L1 data undergo further cloud screening to qualify as Level 1.5 (L1.5). Multiple quality control procedures are applied, including the AOD triplet variability criteria, Angstrom exponent limitations, AOD stability and smoothness checks (Giles et al., 2019). Additional anomaly screenings are applied, including air mass range or minimum daily observations, and result in data being removed from L1.5. Currently, Level 2 (quality-assured) data and sky radiance inversions are not routinely available for in-motion measurements (Torres et al., 2025). Dense aerosol layers, thin cirrus clouds (AOD < 0.03), and obstacles can

introduce erroneous measurements (Eck et al., 2014; Giles et al., 2019).

It is worth noting that photometer AOD uncertainties are estimated at 0.01 for standard channels and 0.02 for UV channels (Eck et al., 1999). Thus, for the wavelengths used in this study (440 and 870 nm), an uncertainty of 0.01 is assumed. Low AOD values have a strong influence on the EAE uncertainty, as the latter depends on the relative errors of AOD. Applying a first order derivatives for error propagation to the simplest EAE(440-870) formula shows that EAE uncertainty is determined

by the product of $[\ln(\lambda_1/\lambda_2)]^{-1}$ (~1.4 for 440/870) and root-sum-square AOD relative errors. For example, assuming AODs of 0.03, meaning relative errors of 33%, the absolute error on EAE is about 0.7.

A comprehensive description of the shipborne photometer system and its data processing procedures is provided by Torres et al. (2025). Moreover, the thoroughly descriptions of AERONET's data quality screenings are presented by Giles et al. (2019).

### 3.2.1 Data quality screening analysis

A multifactor analysis was conducted to evaluate the performance of AERONET's data quality screening procedures for the ship-adapted photometer. The primary focus is on assessing the effect of vessel motion on the passage of measurements through L1 and L1.5 screenings. However, we acknowledge that vessel movement is not the only factor influencing data quality, other environmental and operational parameters, such as cloud coverage, wind speed, temperature and tracking obstructions, also play important roles.

The analysis examines correlations between the proportion of measurements passing L1 and L1.5 screenings and the main influencing factors. To achieve this, conditional counts (i.e., the number of observations meeting specific threshold criteria) were employed to intercompare parameters, including:

- *Apparent wind speed (AWS):* Measured by the main mast weather station.
- *Vessel attitude variability (RMSE$_{HPR}$)*: quantified by the combined RMSE of heading, pitch, and roll (Eq. 4) providing
an estimate of the overall amplitude of vessel motion.

$$RMSE_{HPR} = \sqrt{RMSE_{heading}^2 + RMSE_{pitch}^2 + RMSE_{roll}^2} \qquad (4)$$

- *Cloud coverage:* Derived from lidar measurements and categorized as no clouds, low-to-mid clouds ($\leq 7\ km$), and high-altitude clouds ($> 7\ km$). In particular, the high-altitude clouds are mostly cirrus which do not induce high variations within a triplet measurement; thus, they can pass more easily the triplet criterion.

- *Photometer measurements:* RMSE of the triplet measurement ($RMSE_{DN}$) for L0 to L1 at 440 nm and 870 nm, and zenith angle (ZA) for L1 to L1.5 analysis. Zenith angles were categorized as low (ZA < 30°), mid (30° < ZA < 60°) and high (ZA > 60°).

Thresholds for AWS and $RMSE_{HPR}$ were determined using 25th, 50th and 75th percentiles. Photometer data points were matched with corresponding variables within a 30-second window, consistent with the triplet duration. Data lacking one or more associated metrics were excluded from the analysis. The results of this analysis are presented in Sect. 4.2.2.

After describing the data processing procedures applied to the individual instruments, the next section presents the inversion methodology used to derive aerosol property profiles from synergistic lidar–photometer measurements.

## 3.3 Deriving aerosol properties profiles

The study employs the Klett-Fernald inversion method (Klett, 1981, 1985; Fernald, 1984) to solve the underdetermined lidar equation (Eq. 1). By defining the aerosol extinction-to-backscatter ratio (lidar ratio, LR), the equation is reformulated as a Bernoulli differential equation (Weitkamp, 2005; Speidel and Vogelmann, 2023). The solution depends on boundary conditions defined by the position of an altitude reference zone:

- Backward solution: Reference zone in the far range, dominated by molecular contributions (same as Rayleigh Fit).
- Forward solution: Reference zone near the ground, where aerosol attenuation can be derived or approximated to 1. This solution, however, is mathematically unstable.

In the case of the TRANSAMA campaign, only backward solution is considered, based on daily variations in the range detection limits, as discussed in Sanchez-Barrero et al. (2024). The effective LR (i.e., vertically constant) was iteratively varied to constrain the solution using column-integrated AOD from photometer measurements. As results, vertical profiles of aerosol backscatter ($\beta_{aer}$ in m⁻¹ sr⁻¹) and extinction ($\alpha_{aer}$ in m⁻¹) are derived (Mortier et al., 2013; Popovici et al., 2018). Molecular backscatter and extinction profiles were derived from GDAS (Global Data Assimilation System) data or a tropical standard atmosphere model. GDAS meteorological data is publicly available through NOAA Air resources laboratory (https://www.ready.noaa.gov/READYamet.php, last access: May 9 2025).

### 3.3.1 Uncertainties

Uncertainties are estimated using error propagation from the signal processing by means of first-order derivatives (Russell et al., 1979; Sasano et al., 1985; Kovalev, 1995; Welton and Campbell, 2002; Rocadenbosch et al., 2012; Mortier et al., 2013; Sicard et al., 2020; Sanchez Barrero et al., 2024). Main error sources included overlap function estimation, after-pulse and background noise, while errors in molecular properties are considered negligible. Standard deviations from the overlap correction exceeds 20% below 350 m and decreases to about 5% towards 3 km. Background noise induces maximum error at altitudes where SNR = 1.5, corresponding to ~70% relative error (more details on detection limits are described in Section 3.1.2). Both standard deviation from overlap correction and background noise were propagated from RCS to aerosol retrievals. LR uncertainty was constrained by matching AOD uncertainties ($\pm 0.01$) in the iterative solution. For low-AOD conditions (AOD < 0.05), relative uncertainties increase above 20%, thereby broadening the convergence threshold for LR in the inversion process and leading to larger LR and extinction profile uncertainties. For instance, during the campaign, an AOD of 0.03 corresponded to a LR relative error of approximately 30% (see Section 4.3.2), inducing an additional ~20% error in the extinction profile beyond that propagated from the RCS. The equations used for error calculations are provided in Sanchez-Barrero (2024).

## 4 Results and discussion

Building on the instrumentation and methodologies described above, this section presents the instrumental assessments and the application of lidar and photometer observations. Section 4.1 introduces the local and synoptic atmospheric conditions. Section 4.2 presents the instrumental assessments for both the lidar (Sect. 4.2.1) and the ship-adapted photometer (Sect. 4.2.2), while Sect. 4.3 analyzes the variability of aerosol properties observed during the TRANSAMA campaign, including the atmospheric structure (Sect. 4.3.1) and selected case studies of transatlantic aerosol transport (Sect. 4.3.2).

### 4.1 Local and synoptic scenario

The TRANSAMA campaign comprised three segments: the South Indian Ocean (21–27 April), the South Atlantic Ocean (27 April–11 May), and the North Atlantic Ocean (12–15 May).

An overview of synoptic meteorological conditions, vessel trajectory, and local meteorology is presented in Figure 2. Geopotential height maps at 850 hPa (Figs. 2a–c) reveal large-scale atmospheric dynamics, while temporal series include vessel position and motion (Figs. 2d–e) and meteorological parameters measured from the vessel (Figs. 2f–h) such as temperature (T), relative humidity (RH), apparent wind speed (AWS) and direction (AWD), as well as true wind speed (TWS) and direction (TWD). Data smoothing (3-hour moving averages) was applied for clarity, with boundaries between campaign segments marked by vertical black lines. Geopotential height maps were generated using NOAA/ESRL reanalysis data through

Physical Sciences Laboratory imagery tools (http://psl.noaa.gov/, last access: May 9 2025). Additional meteorological variables, including sea surface temperature, were analyzed but are not showed here for brevity (see Supplements 1).

*South Indian Ocean (21-27 April 2023)*

The campaign's initial segment over the southwestern Indian Ocean was influenced by the Agulhas Current's retroflection, promoting storm activity and rainfall (Lutjeharms and Van Ballegooyen, 1988). A high-pressure system moving eastward supported the passage of a weak low-pressure system (Ndarana et al., 2023) near Madagascar's southern coast (Fig. 2a). Sea surface declined ~0.5 K per degree latitude towards the South (not shown here), with sharp air temperature decrease near South Africa's southern tip (T in Fig. 2f), accompanied by reduced humidity (RH in Fig. 2f) and elevated wind speeds (~20 m/s from

the SSE, TWS in Figs. 2g-h). Vessel motion variability increased ( $RMSE_{HPR}$> 1.3° in Fig. 2e), consistent with synoptic conditions (Fig. 2a).

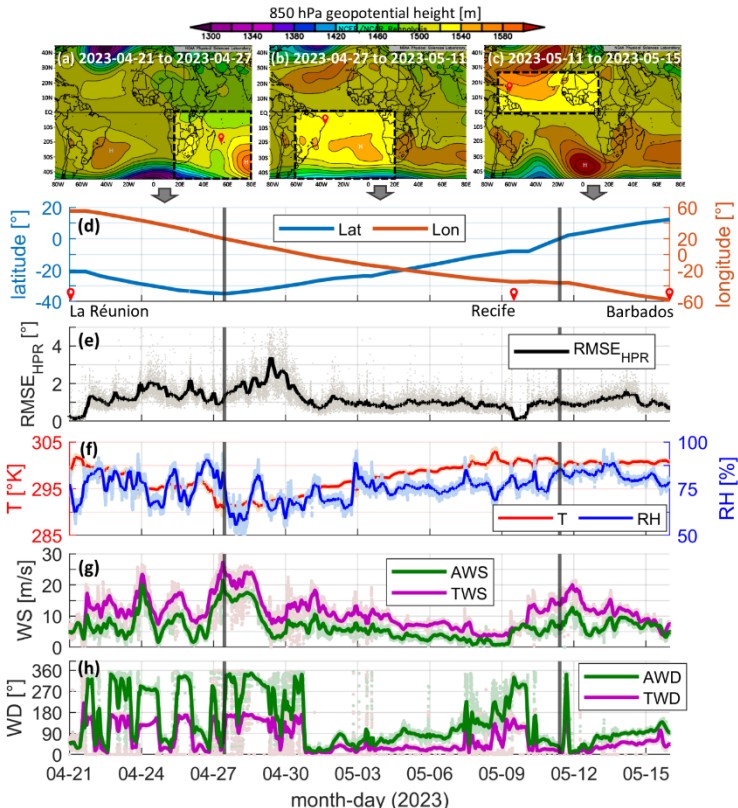

**Figure 2:** Overview of atmospheric and vessel motion data during the TRANSAMA campaign (April–May 2023). Synoptic meteorological conditions are shown using 850 hPa geopotential height maps for (a) April 21–27, (b) April 27–May 11, and (c) May 11–15. Temporal data
for the RV's trajectory are presented in (d) latitude and longitude, (e) the combined RMSE of heading, pitch, and roll (HPRe), (f) temperature (T) and relative humidity (RH), (g) apparent (AWS) and true (TWS) wind speed, and (h) apparent (AWD) and true (TWD) wind direction. Red markers denote ports: La Réunion, Recife, and Barbados. Vertical black lines separate data for the South Indian, South Atlantic, and North Atlantic Oceans. Geopotential height maps were produced using NOAA/ESRL Physical Sciences Laboratory tools (http://psl.noaa.gov/ , last access: May 9 2025).

*South Atlantic Ocean (27 April-11 May 2023)*

The second segment traversed the South Atlantic, marked by interactions between the warm Agulhas and cold Benguela currents, fostering arid conditions along Africa's southwest coast (Shannon and Nelson, 1996). Initial conditions (April 27–30) included elevated wind speeds (TWS>20 m/s IN Fig. 2g) and vessel motion ( $RMSE_{HPR}$~3° in Fig. 2e). Winds eased as high-pressure systems dominated the central South Atlantic (Fig. 2b), producing fair weather under the St. Helena anticyclone (Tyson and Preston-White, 2000). Sea surface temperatures increased (~0.3 K per degree latitude towards the Equator), with TWS < 10 m/s (Fig. 2g) from the north-northeast (Fig. 2e) and $RMSE_{HPR}$ between 0.5–1.3 (Fig. 2e).

*North Atlantic Tropical Ocean (11-15 May 2023)*

The tropical North Atlantic was characterized by stable temperatures (~301 K in Fig. 5f), moderate-to-strong wind speeds (10–18 m/s from the east, TWS in Figs. 2g-h), and moderate $RMSE_{HPR}$ (Fig. 2e). Elevated evaporation rates supported frequent low-level cloud formation, confirmed by lidar observations (see Section 4.3). Moreover, the tropical Atlantic Ocean is influenced by the intertropical convergence zone (ITCZ) shifting northwards during the boreal summer and leading the way for Saharan Air Layer (SAL) transport (Prospero and Carlson, 1972; Barreto et al., 2022).

Although TRANSAMA was a relatively short ship-based campaign, the variability of meteorological conditions, affecting sea state and, consequently, vessel motion, provided a valuable opportunity to evaluate instrument performance under diverse environmental scenarios. Conditions ranged from cloudy, rainy, and rough seas during the transition from the South Indian to South Atlantic Ocean, to calm waters and more clear skies in the mid-South Atlantic, and finally to tropical conditions along the route from Recife (Brazil) to Barbados. These varying conditions enabled us to assess instruments stability and the impact of dynamic sea states on data quality. The next section therefore presents the instrumental assessments for both the lidar and photometer systems.

## 4.2 Instrumental assessments

This section presents the instrumental assessments performed during the campaign. The analyses focus on the performance and data quality of the lidar (Sect. 4.2.1) and ship-adapted photometer (4.2.2) systems under the diverse environmental and motion conditions encountered along the cruise track.

### 4.2.1 Lidar data quality control

The lidar data quality control described in Section 3.1.1 was applied throughout the campaign, and the main results are summarized in Figure 3. Lidar measurements were restricted to international waters; therefore, no profiles were acquired within the territorial waters of Madagascar, South Africa, or Brazil, represented as shaded areas in Fig. 3 (b and d).

During the campaign, a total of 26,215 one-minute profiles were recorded, from which 2,567 Rayleigh fits (Fig. 3a) were performed under clean atmospheric conditions, i.e., defined as cloud-free skies and AOD(532 nm) < 0.1. Coincident

photometer measurements are accounted within a 5-minute threshold. The reference zone $r_{ref}$ was defined between 5 and 7 km a.s.l. (above sea level), and the molecular coefficients were calculated using the tropical standard atmosphere. As expected, fewer Rayleigh fits were performed during the stormy conditions of the Indian–Atlantic Ocean transition (Sect. 4.1) and over the tropical North Atlantic, where clouds were also frequently observed (see more details in Sect. 4.3.1).

From the Rayleigh fits, slope indexes (Fig. 3b and 3c) and the lidar calibration constant (Fig. 3d and 3e) were calculated. This

analysis confirmed the good performance of the lidar system during the campaign, with a slope index showing an overall low bias of 4% relative to the molecular attenuated backscatter profile and a 16% error margin despite the considerable noise induced by solar irradiance. However, the frequent laser attenuation due to sea salt deposition (see Sect. 2.1) prevented a consistent lidar constant for the dataset, with values varying a 30 % relative to the mean ( $C_L = (4.3 \pm 1.4) \times 10^{18} \ photons/s \ m^3 sr$ ).

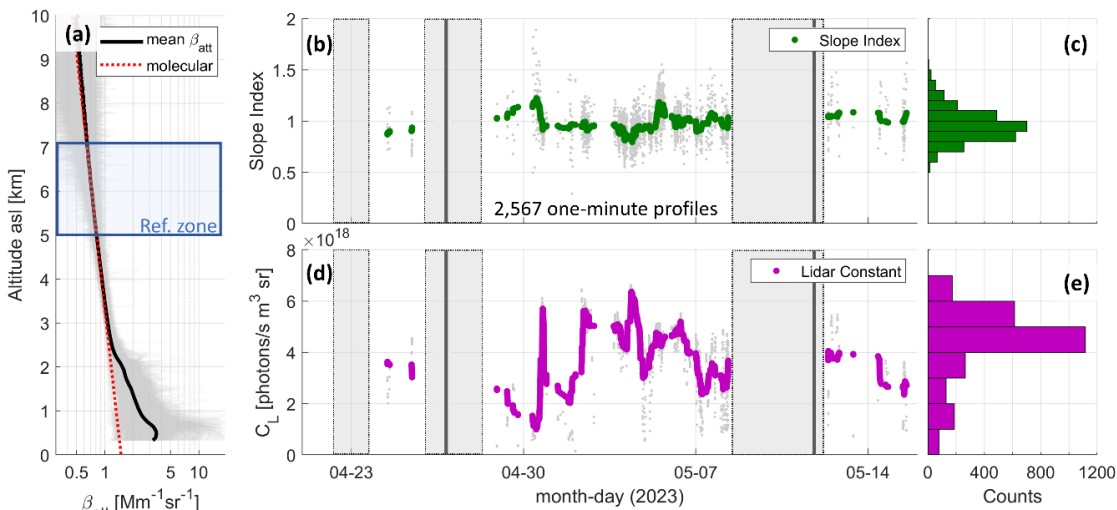

**Figure 3.** Data quality control of lidar profiles. (a) Rayleigh fit procedure was applied on 2,567 profiles (10 % of total) when clean conditions were met and AOD from photometer was available. The slope index temporal series (b) and histogram (c) are presented, along with the derived lidar constant ($C_L$) temporal series (d) and histogram (e). Vertical black lines in (b) and (d) separate data for the South Indian, South Atlantic, and North Atlantic Oceans. The AOD data used for the lidar constant calculation correspond to level 1.5 data of photometer B

(#1243).

The lidar assessments during the campaign served as a reference for the configuration and operation of the newly installed CE376 GP micropulse lidar (working at 532 nm and capable of measuring depolarization) aboard the RV *Marion Dufresne II*, which became operational in October 2025. Similar to the campaign setup of the laser source (Sect. 2.1), the compact CE376 lidar system was installed in a temperature-controlled room on a vibration-isolated and motion-dissipating platform. An extra-

clear glass window was mounted on the roof, ensuring that no mechanical stress was introduced that could alter the polarization response (Sanchez-Barrero et al., 2024). To mitigate the impact of sea-salt deposition, an air-shield chimney was implemented

above the roof window, following the idea behind the design developed for the ship-photometer (Torres et al., 2025). This chimney is easily accessible and permit the regular cleaning of the window. During the campaign a regular cleaning of the window was needed us much as two times a day during stormy days, which we expect to be considerable reduced with the air shield chimney. The system will continuously be operating and mostly will take measurements over the Indian Ocean during the expeditions to the Austral French Territories. More information about the covered region can be found in Tulet et al. (2024), which describes the MAP-IO program.

### 4.2.2 Photometer data screening performance

Building on the methodologies described in Sect. 3.2.1, this section evaluates the operational performance of the two ship-adapted CE318-T photometers deployed during the TRANSAMA campaign. The analysis focuses on the quality of measurements under varying environmental and operational conditions at sea, particularly the impact of vessel motion alongside the roles of wind speed, cloud coverage, and solar zenith angle. Multifactor correlations are used to quantify how these factors affect the proportion of observations passing the L1 and L1.5 screening levels, providing insight into the performance of AERONET protocols in shipborne applications.

Figure 4 summarizes the number of measurements collected at each threshold for both photometers. Panels (a) and (b) show the passage from L0 to L1, while panels (c) and (d) represent the transition from L1 to L1.5. A total of 3,788 and 4,101 triplet measurements (L0) were recorded by photometers A and B, respectively, representing 85–86 % of all observations, including both Sun and Moon observations. Of these, 50–51 % (A) and 55–56 % (B) satisfied the triplet-variance requirement ($RMSE_{DN} \leq 16\%$ at 440 and 870 nm), and 37 % (A) and 40 % (B) subsequently reached L1 after additional checks (as mentioned in Sect. 3.2). The dominant causes of L1 failures were predominantly linked to low-to-mid-level clouds (~30 % of fails with Clouds $\leq$ 7 km) and/or elevated vessel motion (22 % and 20 % fails for A and B with $RMSE_{HPR} > 1.3°$), underscoring the sensitivity of Sun–Moon tracking to ship dynamics.

Among the L1 data (1,409 and 1,642 measurements for A and B), 60 % (A) and 57 % (B), representing ~22 % of all L0 triplets, passed the L1.5 screening (Fig. 4c–d). The L1.5-qualified measurements predominantly occurred under low wind speed (AWS < 4 m s⁻¹), stable vessel attitude ($RMSE_{HPR} \leq 1°$), and mid zenith angles (30°–60°). Cloud contamination was significantly reduced at L1.5; remaining occurrences likely reflect atmospheric inhomogeneity due to differing photometer and lidar viewing angles.

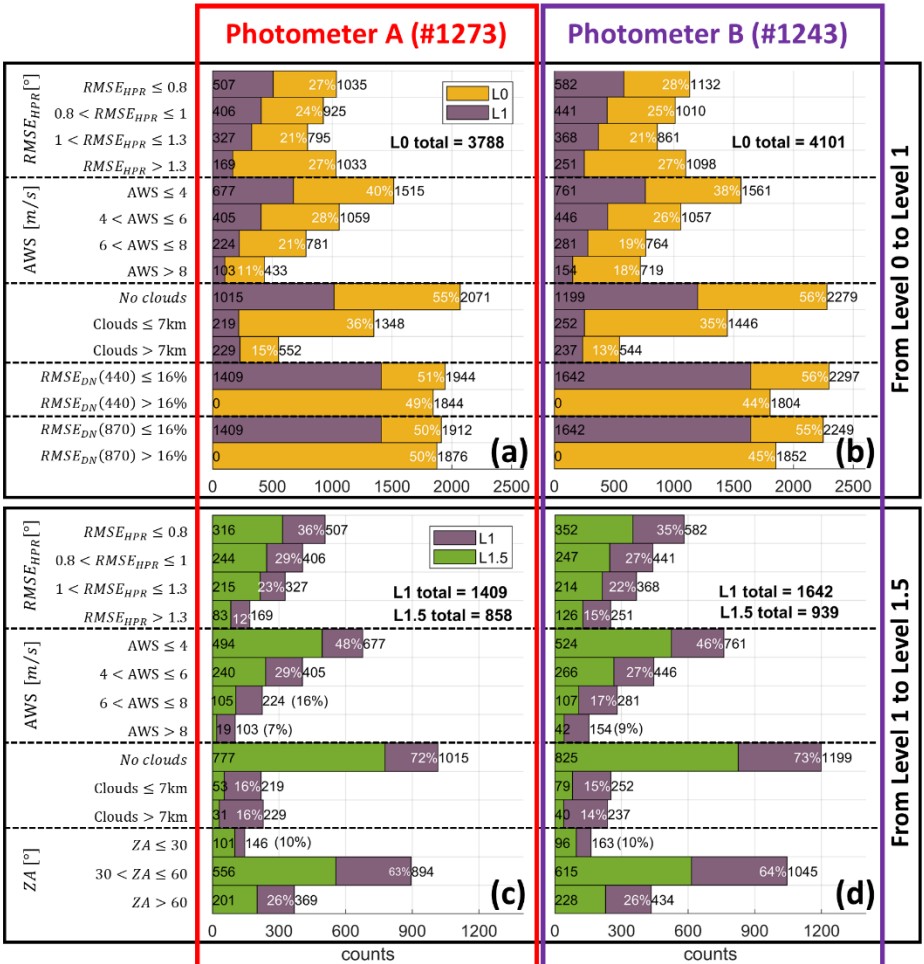

**Figure 4:** Histograms showing the number of photometer measurements falling within the thresholds used in the multifactor analysis. Photometers A (#1273) and B (#1243) are analyzed alongside ancillary parameters: apparent wind speed (AWS), vessel attitude variability ($RMSE_{HPR}$), cloud coverage, triplet RMS errors ($RMSE_{DN}$) and zenith angle (ZA). Panels (a) and (b) display transition from L0 to L1, and panels (c) and (d) from L1 to L1.5.

Severe or rapidly evolving weather conditions affect both the atmosphere and the ocean, but not necessarily simultaneously. Wind forcing induces wave growth, yet changes in wind speed typically precede changes in wave height and sea state. As a result, a rise in wind speed may be followed by a delayed increase in vessel motion (Komen et al., 1994). This decoupling can produce periods in which the atmosphere is relatively calm while the ship still experiences elevated motion (see transition to the South Atlantic Ocean in Fig. 2e,g). To better characterize these interactions, conditional-count matrices were generated to identify how vessel motion interacts with other parameters. Heatmaps in Fig. 5 display conditional counts of $RMSE_{HPR}$ thresholds against other variables at L0 (Fig. 5a–b), L1 (Fig. 5c–d), and L1.5 (Fig. 5e–f). At L0, zenith angle is omitted because it is undefined when the triplet fails tracking stability requirements.

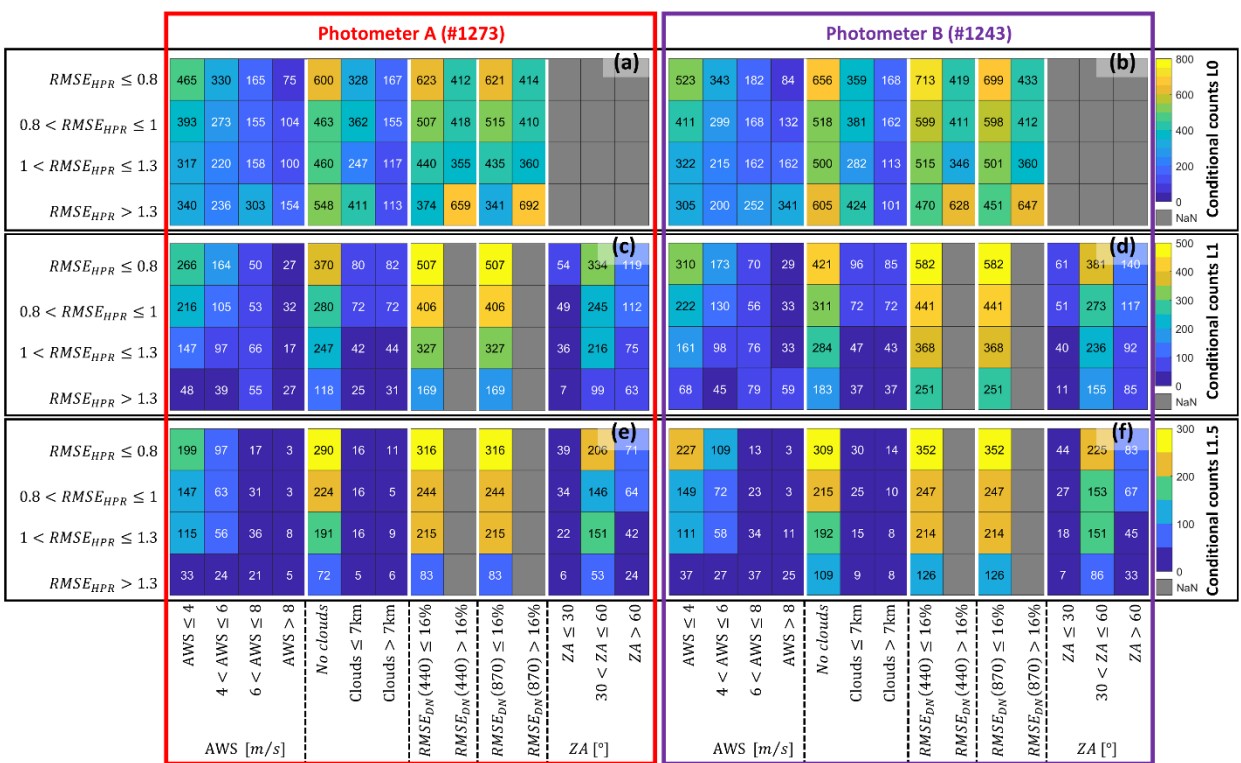

**Figure 5:** Multifactor heatmaps illustrating the relationship between vessel motion ($RMSE_{HPR}$) and other parameters at each processing level for photometers A and B. Panels show conditional counts at L0 (a–b), L1 (c–d), and L1.5 (e–f). At L0, ZA is not included because it is not defined for triplets that fail basic pointing or signal criteria.

Screening success rates decreased considerably under high vessel instability and strong wind speeds. Only 3 % (A) and 7 % (B) of L0 observations taken under $RMSE_{HPR} > 1.3°$ and $AWS > 8\ m/s$ ultimately passed the L1.5 screening (Fig. 5e–f). Moreover, ~17 % (A) and ~15 % (B) of all L0 measurements were linked to large vessel motion and failed the triplet variance criterion (see $RMSE_{HPR} > 1.3°$ and $RMSE_{DN} > 16\%$ in Fig. 5a–b). Among all cases with $RMSE_{HPR} > 1.3°$ (1,033 and 1,098 counts for A and B, see Fig. 4a-b), ~40 % were associated with low-to-mid clouds (Clouds ≤ 7 km), and ~10 % with high

clouds (Clouds> 7 km), both of which may induce higher triplet variance errors at L1 and L1.5 screenings.

Restricting the analysis to cloud-free conditions under $RMSE_{HPR} > 1.3°$, 548 (A) and 605 (B) L0 triplets were identified (~15 % of total data). Of these, only 118 (A) and 183 (B) progressed to L1, and 72 (A) and 109 (B) to L1.5, equivalent to 13% (A) and 18 % (B) of such motion-affected but cloud-free triplets. This illustrates that vessel movement alone, independent of cloud interference, can substantially limit AERONET-quality measurements.

Photometer B collected ~8 % more triplets than photometer A under high-wind conditions, likely due to its more stable installation on the Deck compared to the mast-mounted photometer A (Sect. 2.2). However, photometer B was subject to

occasional obstructions, reducing its L1.5 efficiency across all zenith angles. Small changes in the photometer and installation conditions influence data acquisition performance.

Torres et al. (2025) reported similar behavior from three years of CE318-T operation aboard the Marion Dufresne II, with ~30 % of triplets reaching L1 (compared to 47 % at the nearby Saint-Denis AERONET site), and ~70 % of L1 data reaching L1.5 (vs. 80 % at Saint-Denis). The TRANSAMA campaign achieved a higher performance in terms of L0 to L1 passage (~37–40 %), likely due to the prevalence of calmer mid-South Atlantic conditions compared to the rough seas encountered during the Antarctic lands missions. In contrast, the fraction of L1 data advancing to L1.5 (~60 %) was ~10 % lower than the long-term performance reported by Torres et al. (2025), potentially reflecting short-lived tracking disturbances caused by small vessel motion and thin cirrus or transient low clouds in the tropics. These comparisons underscore that while vessel motion is a key limiting factor, differences in screening efficiency cannot be attributed solely to ship dynamics; atmospheric variability also plays a central role. Moreover, experience aboard smaller ships shows that not only the amplitude but also the *frequency* of vessel motion may critically affect tracking stability. For faster and smaller ships, we have considered the use of stabilizing platforms to improve the data acquisition (see discussion in Torres et al. (2025)).

Thus, to better quantify the detection limits of ship-borne photometers under vessel motion, a reference photometer is now being operated alongside a second unit mounted on a motion-simulation hexapod platform. This configuration enables controlled experiments to isolate how specific motion modes (i.e., varying amplitude and frequency of pitch and roll) affect data quality and sky-measurement geometry (results will be analyzed in future studies). These ongoing efforts are expected to yield quantitative thresholds for operational performance, guide future improvements to ship-adapted photometer systems across different ship classes, and support the evaluation of potential improvements on data-quality screening protocols.

Building on the lidar and photometer instrumental assessments, the following section examines how the combined dataset captured the aerosol variability encountered throughout the TRANSAMA transect.

### 4.3 Aerosol variability during TRANSAMA

Aerosol properties were assessed using a combination of lidar and photometer data. Figure 6 illustrates the spatio-temporal variability in lidar and photometer observations. NRB profiles (Fig. 6a) highlight aerosol and cloud distributions, while photometer observations of AOD at 440 nm (Fig. 6b) and EAE at 440–870 nm (Fig. 6c) provide aerosol size and concentration insights. Both L1 and L1.5 levels of photometer observations are plotted. The campaign encountered clean marine conditions with AOD at 440 nm of 0.08 ± 0.04 and EAE at 440–870 nm of 0.5 ± 0.2. As mentioned in Sect. 4.2.1, active sensing restrictions in Madagascar, South Africa, and Brazilian territorial waters limited lidar observations; however, passive photometer measurements continued with exception in Brazil's coastal zones.

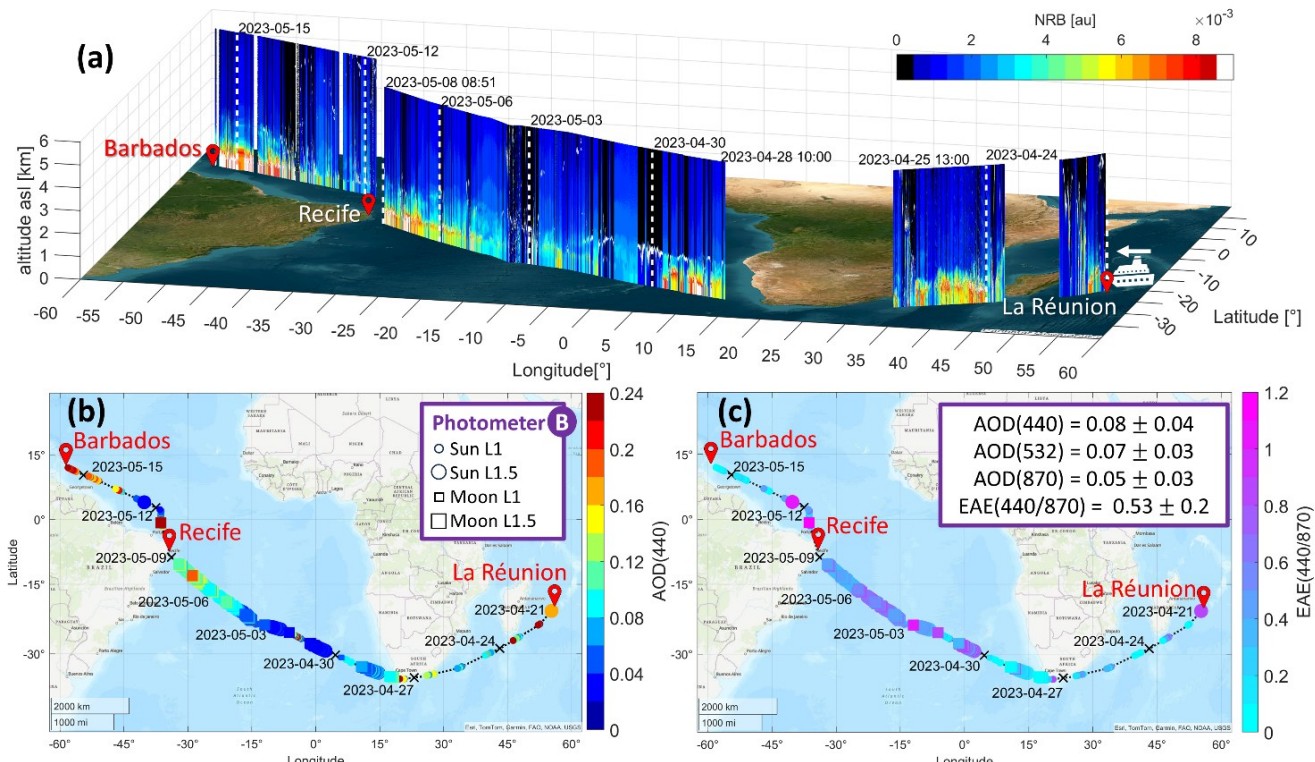

**Figure 6:** Spatio-temporal variability of aerosol properties during the TRANSAMA campaign (21 April–15 May 2023) aboard the RV *Marion Dufresne II*. Measurements were conducted along the route from La Réunion Island to Barbados. (a) 3D variation of NRB at 532 nm from lidar measurements overlaid on a true-color image of the covered regions. (b) AOD at 440 nm and (c) EAE at 440/870 nm derived from photometer B observations, displayed on topographic maps. Photometer data include L1 and L1.5 solar and lunar observations. Red pins mark the ports at Le Port (La Réunion), Recife (Brazil), and Bridgetown (Barbados). Image basemaps provided by © Earthstar Geographics | Esri using MATLAB®.

*South Indian Ocean (21-27 April 2023)*

In agreement with the meteorological conditions (Sect. 4.1), persistent low clouds (see lidar NRB profiles, Fig. 6a) and rainfall reduced photometer data acquisition (Figs. 6b-c). L1.5 photometer observations were primarily obtained during docking at La Réunion (April 21), where AOD(440) values reached 0.16, and EAE(440–870) indicated presence of fine aerosols, likely linked to urban emissions and volcanic degassing (volcano reported active that day).

*South Atlantic Ocean (27 April-11 May 2023)*

The central South Atlantic provided the most favorable observing conditions of the campaign. Both lidar and photometer were able to measure continuously during this segment. Photometer AOD(440) values ranged between 0.03-0.1 (Fig. 6b), with EAE(440/870) ranging 0.4–0.8 (Fig. 6c), yelling pristine marine conditions (Hamilton et al., 2014), and consistent with clean

marine aerosol classifications (Kumar et al., 2017; Ranaivombola et al., 2023). The extended range of EAE values are related to the absolute errors (~0.7) induced by the low AOD values (Sect. 3.2). On the other hand, lidar observations revealed transient clouds within the first 2 km and thin aerosol layers above the MBL (more details discussed in following sections). Near Brazilian territorial waters, AOD increased to 0.15, and EAE rose to 0.6–0.9, suggesting coastal-urban influences within the dominant marine conditions.

*North Atlantic Ocean (11-15 May 2023)*

Frequent low-level clouds and recurring thin cirrus reduced the number of photometer samples passing L1.5 screening. Lidar observations confirm persistent cloudy conditions throughout this segment. A SAL intrusion event was indicated by external dust forecasts (Barcelona Dust Regional Center, https://dust.aemet.es/, last access: May 09 2025), but cloud cover prevented detailed analysis of the aerosol vertical structure during the event. During this period, only L1 photometer data were available, consistent with the lidar detection of both low clouds and cirrus. The stronger AOD values recorded near Barbados (~0.2) did not pass L1.5 screening; however, when considered together with the low EAE values (<0.2), these observations are consistent with the possible presence of dust. Although these measurements are not quality-assured, the temporal continuity of the AOD and EAE suggests that some valid dust-affected observations may have been incorrectly discarded by the screening protocols. This outcome further highlights the need to evaluate whether dedicated ship-photometer AERONET protocols could improve data retention under marine conditions where motion and transient cloud contamination frequently challenge standard screening criteria.

This overview lays the foundation for a detailed examination of atmospheric structure, focusing on vertical aerosol distributions and insight into interaction of clouds and aerosols, as discussed in the next section.

### 4.3.1 Atmospheric structure

Lidar measurements accounted for 73% of the total navigation time (25 days) from La Réunion to Barbados, generating 26,215 one-minute profiles. Among these, 53 % were flagged as cloud-contaminated. Figure 7a illustrates histograms of cloud and aerosol layer presence across 150 m altitude bins, determined using the gradient method described in Section 3.1.3.

Cloud coverage extended over a broad altitude range, with cirrus clouds reaching above 7 km and up to the tropopause (~17 km near the equator). High-altitude cirrus clouds (>13 km) were observed exclusively within ±5° of the equator, where the tropopause is higher compared to subtropical regions. Cloud occurrence was highest (>10%) below 2 km, coinciding with increased aerosol concentrations. However, cloud presence may be slightly underestimated due to the thresholds applied during automatic detection. Transient clouds in the lower troposphere and high cirrus clouds masked by noise presented challenges for detection, impacting the subsequent inversion.

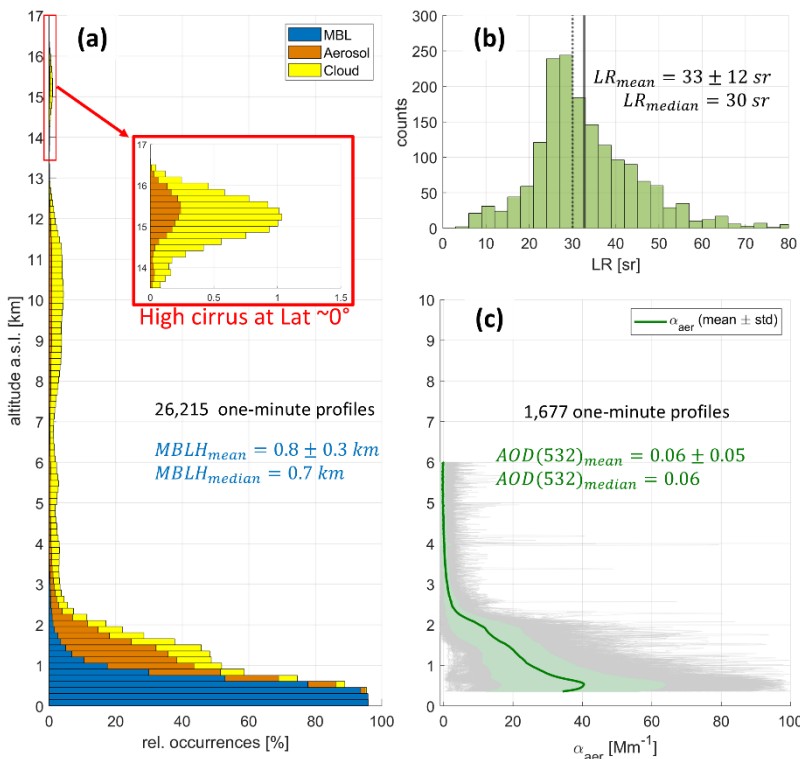

**Figure 7:** Aerosol vertical distribution derived from lidar and photometer. (a) The occurrence of the MBL, aerosol and cloud layers are presented along with the aerosol (b) effective LR and (c) extinction coefficient profiles.

The mean (median) MBL top height (MBLH), determined for the campaign, was $0.8 \pm 0.3$ km (0.7 km), showing minimal daytime growth. These findings align with prior studies suggesting MBL heights over remote oceans, coastal regions, and islands typically range between 0.5 km and 1.5 km (Flamant et al., 1997; Bates et al., 2002; Cosma-Averseng et al., 2003; Luo et al., 2014, 2016; Bohlmann et al., 2018; Díaz et al., 2019; Barreto et al., 2022). Unlike the daily evolution observed over land, the oceanic MBL displayed limited daily growth when the vessel was distant from coastlines, primarily due to the high

heat capacity of water (Stull, 1988; Oke, 2009). As noted in Section 4.1, temperatures showed latitudinal increases towards equator rather than daily variations.

Comparatively, Bohlmann et al. (2018) reported lower MBLH (300–500 m) in a two-day case study near Cape Town using multi-wavelength Raman lidar aboard the Polarstern vessel. This discrepancy might reflect the impact of colder waters in the Cape Town region on MBLH growth. Additionally, the lidar system's blind zone (~350 m, Section 3.1.2) limited detailed

analysis closer to sea level, underscoring the need of enhanced lidar data acquisition near the surface.

Quality-assured one-minute lidar profiles (Section 4.2.1) underwent an inversion procedure, followed by filtering to exclude profiles with LR exceeding 100 sr or negative extinction coefficients within the first 1.5 km. Campaign data returned 1,677 profiles under clear sky and stable to moderate vessel conditions, mostly over the South Atlantic (Section 4.1). To derive

effective LR (Fig. 7b) and extinction profiles (Fig. 7c), stratospheric aerosol contributions from the Hunga underwater volcano eruption (Boichu et al., 2023) were removed. This was done by subtracting the stratospheric AOD from the total column-integrated AOD used to constrain the inversion. By April 2023, Sicard et al. (2025) reported a stabilized aerosol plume at 21°S between 18.5 and 23.5 km, contributing a stratospheric AOD of 0.012 at 355 nm. Accounting for this contribution (estimated to ~0.01 at 532 nm), the derived mean (median) LR was 33 ± 12 sr (30 sr), with a 27% overestimation (LR = 42±16 sr) when neglecting the stratospheric impact.

While photometer observations identified clean marine aerosols, LR values aligned with studies on contaminated and dried marine aerosols (Duflot et al., 2011; Burton et al., 2013; Floutsi et al., 2023). The mean extinction profile ($10 - 40\ Mm^{-1}$) extended up to 2 km is consistent with low aerosol loadings and aerosol presence above the MBL. Luo et al. (2016) identified boundary layer decoupling over the remote Pacific, modulated by inversion strength. This decoupling leads to vertical stratification of aerosols, with distinct layers corresponding to the internal mixing layer and the boundary layer top. The resulting structure influences cloud-top properties and cloud formation. Similarly, our observations showed increased aerosol concentrations above the MBL, coinciding with higher cloud occurrence and consistent with stratification scenarios described by Luo et al. (2016).

Having analyzed the atmospheric structure, following section examine the implications of these findings on transatlantic aerosol transport.

### 4.3.2 Transatlantic transport

Despite the clean conditions identified in the South Atlantic Ocean, lidar consistently detected persistent thin aerosol plumes above the MBL throughout the route. Under the presence of these layers, profiles exhibited LR values (33 ± 12 sr) slightly exceeding those typical of pure marine aerosols (Sect. 4.3.1), suggesting contributions from transported aerosols. While marine aerosol production usually increases weakly with wind speed (e.g., Smirnov et al., 2012), recent analyses (Sun et al., 2024) show that this effect is mostly confined within the MBL. In our dataset, decreasing wind speed coincided with increasing AOD (not shown here), suggesting that the negative correlation arises from vertical distribution and transport processes rather than a contradiction of wind-driven aerosol production (Smirnov et al., 2012; Sun et al., 2024).

Observations from May 4–6, 2023, captured height-time variations of NRB at 532 nm (Fig. 8a) and photometer AOD and EAE (Fig. 8b), benefiting from reduced cloud presence. Notably, AOD increased from ~0.04 to ~0.1 after 12:00 UT on May 4, with EAE(440/870) between 0.5 and 0.75, indicating mixed aerosol sizes.

To improve lidar SNR, six-time intervals (T1–T6 in dashed white lines on Fig. 8a) were selected for profile averaging (15–30 min). Corresponding back-trajectories were generated using HYSPLIT model (Hybrid Single-Particle Lagrangian Integrated Trajectory; https://www.arl.noaa.gov/hysplit/, last access: May 8 2025), employing GDAS 1-degree resolution meteorological

data and tracing aerosol paths up to 2 km altitude over 12 days. These trajectories (Fig. 8c) were overlaid on MODIS (Moderate

Resolution Imaging Spectroradiometer) Aqua/Terra true-color imagery, with thermal anomalies marked as orange points (image generated through NASA Worldview; https://worldview.earthdata.nasa.gov/, last access: May 8 2025). Observations from AERONET sites on St. Helena (elevated, 400 m a.s.l.) and Ascension Islands provided additional context (Fig. 8d). Notably, St. Helena was within the generated back-trajectory paths, while Ascension Island lay outside.

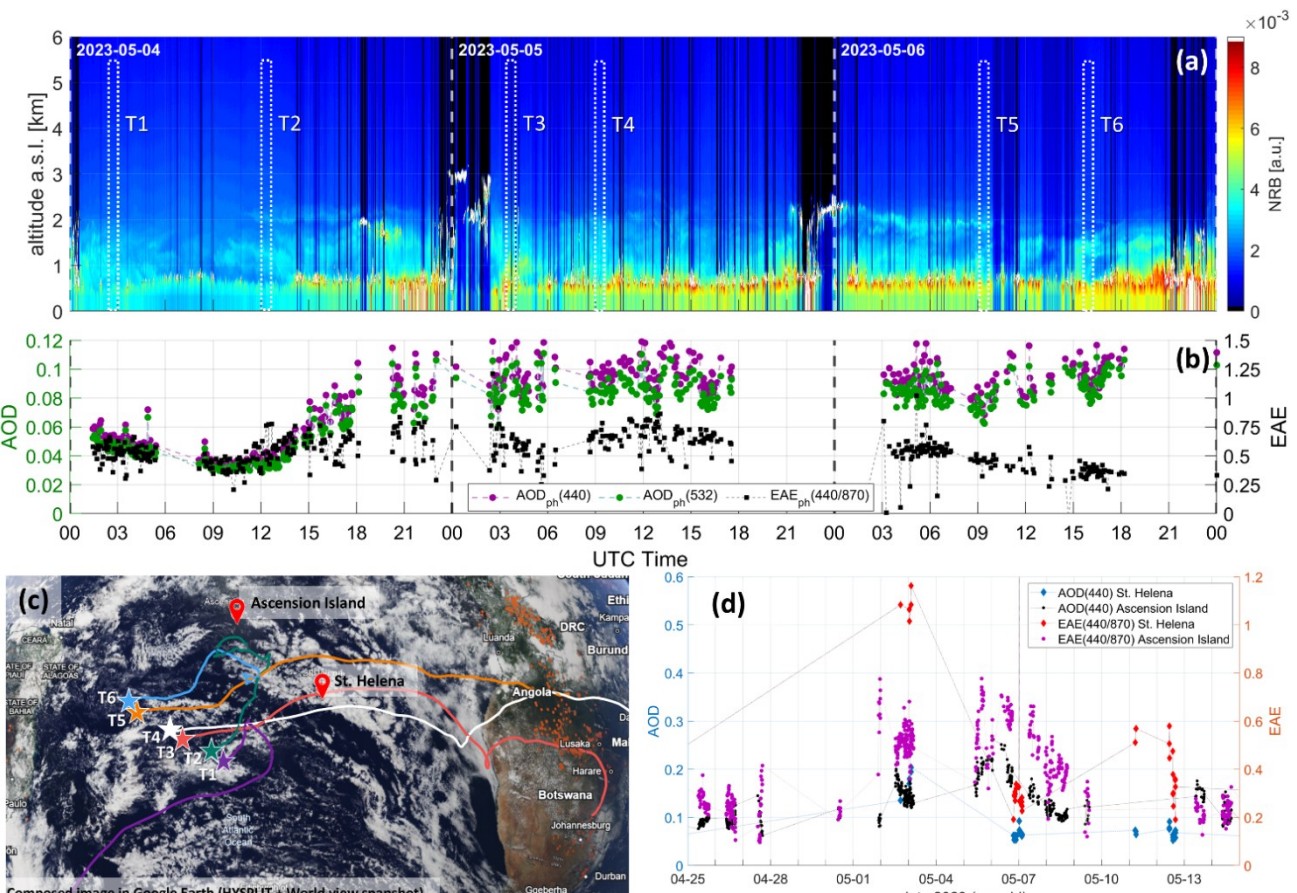

**Figure 8**: (a) Height-time variations of NRB at 532 nm and (b) temporal sun/moon photometer observations during May 4–6, 2023, are shown. (c) HYSPLIT back trajectories for selected time thresholds (T1–T6) overlaid on MODIS Aqua/Terra true-color imagery from NASA Worldview (composed image using ©Google Earth) and (d) temporal series of AOD and EAE from two AERONET sites on islands (St. Helena and Ascension). Sunrise and sunset occurred at ~07:00 UT and ~20:00 UT, respectively. The time thresholds considered for the back-trajectories are indicated by dashed white lines in (a). Photometer data is L1.5 for Marion Dufresne photometer A, sun and moon measurements from AERONET island sites are L1.5 and L2 respectively.


Figure 9 presents the backscatter and extinction profiles for the selected time intervals, offering insights into aerosol mixing dynamics. Similarly, stratospheric AOD was taken into account for in the inversion (Sect. 4.2). While prior studies (Luo et al., 2016; Sun et al., 2024) associated MBL decoupling with vertically stratified marine aerosol layers, our observations suggest a more complex mixing process. Aerosol plumes above the MBL, embedded within an extended mixing layer (reaching up to

2 km), showed varying contributions from marine and transported aerosols. Extinction and backscatter coefficients were consistently higher within the MBL, indicating dominant AOD contributions from marine sources. For example, aerosol layers in datasets T3 and T5 showed extinction peaks ($21 \pm 5$ Mm⁻¹) less than 50% of MBL values ($60 \pm 15$ Mm⁻¹), with lower LR values ($19 \pm 2\ sr$ and $25 \pm 5\ sr$), closer to typical clean marine conditions. In contrast, other profiles exhibited higher LR values (up to 40 sr), reflecting transported aerosol influence.

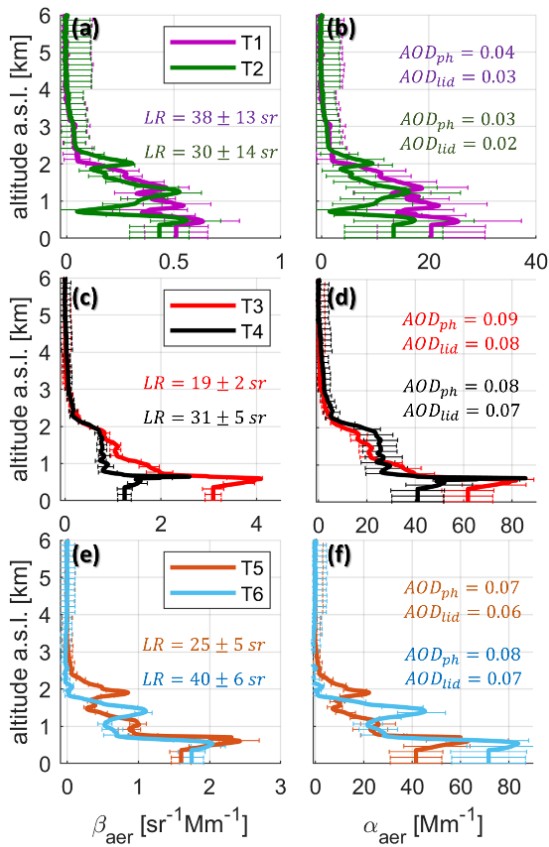


**Figure 9:** Aerosol properties of averaged profiles during selected time intervals (T1–T6 in Fig. 8a) are shown. Backscatter (a, c, e) and extinction (b, d, f) coefficients profiles are presented. Molecular contributions were estimated using GDAS-1degree meteorological data.

Uncertainties were larger for low-AOD profiles (T1–T2, AOD <0.05), with relative errors above 30% for LR and >20% for backscatter coefficients below 3 km, leading to extinction errors >50%. For higher AOD loadings (T3–T6, 0.06–0.08), LR and 590 backscatter errors dropped below 16% and 15%, respectively, yielding extinction uncertainties around 30%. Despite these challenges, the profiles provide valuable information on aerosol optical properties.

Back-trajectories revealed distinct aerosol origins. For datasets T3, T4, and T5 (Fig. 8c), aerosols traveled ~4,500 km over 12 days, originating from wildfire regions in southeastern Angola, southwestern Zambia, and northern Botswana, passing through urban areas. Thermal anomalies identified in these regions (Fig. 8c) suggested fire activity prior the well-known wildfire season

in South African region (Ranaivombola et al., 2023 and references therein). Conversely, datasets T1, T2, and T6 suggested air masses circulating over the ocean for ~4 days with potential contributions from the mixed residual biomass-burning-urban emissions. Supporting data from AERONET sites underscored the increase of AOD at 440 nm during the first days of May and the role of fine aerosols. For instance, St. Helena's site (400 m a.s.l.) recorded increased EAE(440/870) above 1 and AOD(440) exceeding 0.1 on May 3, indicative of fine aerosol presence. CALIOP (Cloud–Aerosol Lidar with Orthogonal

Polarization) satellite-based lidar observations from the CALIPSO (Cloud–Aerosol Lidar and Infrared Pathfinder Satellite Observations) mission also detected similar elevated aerosol layers on 5 May, classified as biomass-burning aerosol (not showed here, see Supplements 2).

While the influence of residual biomass-burning-urban aerosols was modest (extinction peak <30 Mm$^{-1}$), the persistence of thin aerosol layers throughout the route highlights the significant impact of continental emissions on mid-ocean conditions.

These transported aerosols likely contribute to oceanic cloud formation, evidenced by lidar observations of cloud development within the first 2 km in altitude.

## 5  Conclusions and perspectives

This study highlights the unique insights gained from the TRANSAMA campaign, by using a synergistic combination of micropulse lidar and ship-adapted photometers to investigate aerosol properties in marine environments. The campaign

covered the South Indian and Atlantic Oceans, offering the following key conclusions:

- The coupled micropulse lidar-photometer system demonstrated robust performance in challenging shipborne conditions, effectively capturing vertically resolved aerosol properties despite environmental constraints such as sea spray deposition (for lidar) and sharp vessel motion (for the photometers). This achievement highlights the feasibility of deploying compact, automated remote sensing instruments for continuous aerosol monitoring in marine

environments.

- The vertical profiles derived from lidar data revealed persistent thin aerosol layers above the MBL, even in conditions that might otherwise be misclassified as pristine. These layers predominantly observed over the South Atlantic Ocean, were attributed to long-range transport of residual biomass-burning-urban aerosols from Southern Africa. Back-trajectory analyses and LR values ($33 \pm 12\ sr$) corroborated these findings, aligning with previous studies on

contaminated marine aerosols. These results emphasize the significant impact of continental emissions on remote oceanic regions.

- The MBL top height was determined to be $0.8 \pm 0.3$ km, showing minimal daily evolution, consistent with the thermal inertia of oceanic surfaces. Frequent cloud coverage was observed in lidar profiles, often in association with aerosol presence within and above the MBL. These findings suggest complex aerosol-cloud interactions at low aerosol

 loadings, emphasizing the importance of integrated observation networks to unravel aerosol transport mechanisms and their role in cloud formation in pristine regions.

Building on the instrumental assessments presented here, a CIMEL CE376 depolarization lidar was installed aboard the Marion Dufresne in October 2025 within the ACTRIS-FR/OBS4CLIM project. Together with the shipborne photometer, this system will enable regular monitoring of the southwestern Indian Ocean during the vessel's recurrent missions to the French Southern and Antarctic Territories (TAAF). Future developments include the implementation of a dedicated data processing chain for mobile lidar–photometer observations at the AERIS Data and Service Center, contributing to the ACTRIS-FR exploratory platforms.

Further ship-adapted photometer assessments will be carried out to quantify the detection limits imposed by vessel motion and to support future improvements in instrument design and data-quality screening protocols. The MAP-IO program and complementary efforts to equip research vessels with advanced Sun–sky–lunar photometers, including the Gaia Blu RV operating in the Mediterranean, represent key steps toward establishing an extended oceanic aerosol network. Integrating vertically resolved lidar measurements within this network will substantially enhance our understanding of aerosol transport and dynamics, with direct relevance for climate modelling and satellite calibration and validation activities.

**Data availability**

Data from photometer land sites are available at AERONET website (https://aeronet.gsfc.nasa.gov, last access: 25 April 2025). The MODIS imagery products are available at NASA Worldview (https://worldview.earthdata.nasa.gov, last access: May 8 2025). HYSPLIT back-trajectories and GDAS-1degree meteorological data were generated using READY (Real-time Environmental Applications and Display sYstem) NOAA ARL website (https://www.ready.noaa.gov, last access: May 8 2025). Marion Dufresne data used in this paper, including meteorological, attitude, lidar and ship-adapted photometer data, are available upon request to the corresponding author.

**Authors contributions**

MFSB prepared the figures and wrote the manuscript. PG, LB, BT, IEP and MS participated in the discussion of methodology and results. BT, LB, PG and GD lead the ship-adapted photometer developments. TP, LB, IEP and MFSB supported instrumental assessments of lidar. PG, CS, LB, VBR, MS and MFSB contributed to the organization and supported the installation and assessments of instruments aboard the *Marion Dufresne II* RV during TRANSAMA campaign. FD, RDF, TP and LB assured the data transmission to the LOA server and NRT data treatment.

## Competing interests

The authors declare no conflict of interest.

## Acknowledgements

This work was supported by the ESA-funded project QA4EO (Quality Assurance Framework for Earth Observation), the EUMETSAT-funded project FRM4AER (Fiducial Reference Measurements for Copernicus Aerosol Product Cal/Val Activities), the Horizon Europe European Research Council (project REALISTIC, grant no. 101086690), the Agence Nationale de la Recherche (project OBS4CLIM, grant no. ANR-21-ESRE-0013) and CNES through the projects EECLAT, AOS, and EXTRA-SAT.

MAP-IO was funded by the European Union through the ERDF programme, the University of La Réunion, the SGAR-Réunion, the Région Réunion, the CNRS, IFREMER, and the Flotte Océanographique Française. The technical developments of the shipborne photometer are part of the joint laboratory AGORA-LAB (a collaboration between LOA and CIMEL Electronique).

The authors highly acknowledge the support of TAAF, IFREMER, LDAS, and GENAVIR for their assistance in the installation and maintenance of scientific instruments aboard the Marion Dufresne II RV. Special thanks are also extended to
the technical teams of LACy and OSU-R for their efforts in data acquisition and instrument maintenance. The authors also thank the principal investigators (Pawan Gupta and Elena Lind) effort in establishing and maintaining St. Helena and Ascension AERONET sites.

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
