# Peer review of "Aerosol variability over oceans using micro-pulse lidar and photometer: Insights from TRANSAMA ship-based campaign"

_EGUsphere, 2025_

## Referee Comment (RC1)

**Referee comment on "Aerosol variability over oceans using micro-pulse lidar and photometer: Insights from TRANSAMA ship-based campaign" by Sanchez-Barrero et al.**

Anonymous Referee

This is a short-term campaign (April–May 2023) that could prove useful for improving current techniques related to photometers installed on board mobile platforms. However, this task seems to have already been addressed in a previous article (Torres et al., 2025), which made use of three years of measurements from the same platform described in the present work, and which, in fact, includes the period covered by this article.

The main objective of this article, namely to improve our understanding of aerosol properties in a marine environment, appears rather ambitious given the very limited measurement period employed. Nevertheless, this article can provide valuable experience for the deployment of aerosol observations on oceanic platforms, which are essential to enhance our knowledge of their global-scale concentrations and of their possible interactions within cloud formation processes and other climatic processes.

Therefore, this is a paper that provides necessary and useful information for the scientific community, is well-written, and is supported by high-quality data, perfectly aligning with the scientific objective of this journal. I thus recommend its publication in AMT with some scientific and technical comments.

**General comments:**

I wonder whether the main objective of this study is to increase our knowledge of aerosols in marine regions. From my point of view, the importance of this article lies in the simultaneous measurement on board a vessel of a photometer (already presented in a previous article) and a lidar (a clear novelty introduced in this study). At first, I understood that this work was intended to lay the foundations for mobile measurements with both instruments, but after completing the reading I am not entirely sure what Section 3 of the article actually contributes. I do not understand Figures 3 and 4, and initially I thought that new quality-control level criteria adapted to ship-based measurements were being introduced. Later I realized that this was not the case. In short, I do not clearly understand the main objective of this article nor the information presented in Section 3.

What is clear, however, is the usefulness of the synergistic information presented in this article for performing real-time atmospheric monitoring during the vessel transects. A clear example is the information shown in Figure 6. Section 4 presents relevant and useful information for continuous monitoring, but it perhaps includes too many variables and explanations, which may cause the reader to lose track of the sequential objective of the paper.

**Scientific/Technical comments:**

Paragraph 45-56: In this paragraph, the authors describe the current efforts to fill the observational gaps that still exist in the study of marine aerosols. They list a series of interesting campaigns and technical efforts that form part of the state of the art. However, and somewhat surprisingly, they do not include the scientific article that reports on the three-year dataset obtained with the mobile device designed for the same vessel by the same authors. It is not until line 78 that this article is referenced. Please revise the state of art about marine photometric campaigns.

Line 60: Forest fires are not included here?

Line 63: Is there any example of mobile Raman lidars to be added here?

Line 116: Please include here the reference to Giles et al. (2019).

Line 122: This sentence seems incomplete. Maybe it has sense if "which" is replace by "from"?

Line 132: The use of the connector "nevertheless" in this sentence appears somewhat confusing.

Line 132: Regarding the PLASMA measurements, although they are not included in the current manuscript, were they conducted successfully? Do the authors plan to report them in a future publication? In my view, if the availability of PLASMA measurements is mentioned in the text, it would be helpful to provide some additional details about them to inform the reader. However, this is merely a suggestion for clarity and completeness.

Line 141: If the objective of this paper is to lay the foundations for lidar measurements on board a mobile platform, I believe it is necessary to provide details on how frequently cleaning should be performed, as well as additional technical information that could be useful for other similar experiences.

Line 204: Is there any difference compared to the values found "on the ground"?

Line 236: Is Eck et al. 2014 a correct reference to state here the definition of AERONET Level 1 algorithm? In this regard, I must admit that I find this paragraph generally confusing. Perhaps it stems from a misunderstanding on my part, but I understand here that AERONET Level 1 is based solely on triplet filtering according to a threshold. However, later on, in line 262, it is stated that there are additional requirements for a measurement to progress from Level 0 to Level 1, according to Giles et al. (2019). Could the authors please clarify this point?

Line 240: The authors clarify that Level 2 is not yet available for mobile data. Consequently, the following sentence regarding the possible presence of aerosol or cloud layers suggests that, due to the absence of Level 2, such structures may result in erroneous

measurements. Additionally, I wonder whether the term "uncertainty" is the most appropriate in this context, or if another term might better convey the intended meaning.

Line 250: Why is "Ve" used as the acronym for triplet measurements? The same for HPRe.

Sect. 3.2.1: At this point, this referee was expecting the definition of specific quality levels for ship-based photometric measurements. WS, HPRe, CC, and Ve are introduced, and Figures 3 and 4 are presented. I have many questions regarding this.

- In line 259, it is stated that these figures illustrate the filtering process. However, it seems that they actually provide an insight of the real conditions under which the AERONET criterion filters the data, is that correct? Or is a new process being introduced here?
- How are the 37% and 40% values in line 261 obtained? Are they averages of all data classified as subset frequency data L1?
- I may not have fully understood Figures 3 and 4 (a) and (b), but shouldn't the sum of all bars for each variable equal the total number of L0 and L1 data points? For example, in Figure 3(a), the total L0 is 3786. The sum of counts by WS is 3786, which is as expected. However, for HPRe it is 3737, and for Ve it is 7572. Could the authors please clarify this point?
- Subset frequency data refers to the amount of data passing and specific screening?
- In summary, I find it difficult to quickly and effectively interpret the information presented in these figures. Has the information provided here been used to offer measurement guidance or advice that could be useful for improving the filtering criteria designed for on-ground instrumentation? I do not see any other reference to these numbers throughout the text. I leave it to the authors to decide whether this information is sufficiently relevant to be included in the manuscript, or whether it should rather be summarized or removed from the final version.

Line 284: Which reference is this estimate based on?

Line 329: Why is "clean" placed in quotation marks? I believe it would be more appropriate to provide a bibliographic reference for such conditions.

Line 426: It would be very illustrative to include the value obtained here again.

Line 428: The authors found a negative correlation between AOD and WS. Could they elaborate further on the hypothesis regarding the origin of this unexpected anticorrelation?

References: The reference to Torres et al. (202) appears to be incomplete.

---

## Referee Comment (RC2)

[revised manuscript text omitted]

*The primary aim of the study is to demonstrate the benefits of combining LiDAR and photometer technology on vessels.*

The primary objective of this study is to enhance our understanding of aerosol properties in marine environments through the synergistic application of micropulse lidar and ship-adapted photometers during the TRANSAMA campaign (April–May 2023). This work focuses on characterizing the vertical distribution of aerosols and clouds across the South Indian and Atlantic Oceans, with particular attention to the influence of long-range transported continental aerosol plumes on remote oceanic regions. The study also aims to validate and assess the performance of an integrated lidar-photometer system under challenging conditions, demonstrating its capability (and limitations) for continuous and autonomous observations. By bridging observational gaps over oceans, this approach contributes to ongoing discussions on aerosol-climate interactions, satellite calibration and validation, and emphasizes the need for comprehensive observational strategies in marine environments.

*There is insufficient data for a study of aerosols above the oceans. This is an example of a potential application.*

This work presents the description and setup of the remote sensing instrumentation used during the TRANSAMA campaign in Section 2. The data processing and quality control for both lidar and photometer are presented in Section 3. Results and discussion of the observed aerosol properties in maritime environment are presented in Section 4, highlighting atmospheric structure and transatlantic transport.

**2 Remote sensing instrumentation**

*This paragraph should be placed at the end of the introduction.*

This section outlines the mobile remote sensing instruments used in this study. Subsection 2.1 describes the CE370 lidar, while subsection 2.2 describes the CE318-T photometer. The instrumentation setup during the TRANSAMA campaign is detailed in Subsection 2.3.

*Measurements have been performed using passive and active remote sensing. The raw data were transmitted...*

**2.1 Single-wavelength lidar**

The ***CE370 lidar*** is an eye-safe, single-wavelength micro-pulse elastic lidar operating at 532 $nm$ with a 20 $\mu J$ pulse energy at a 4.7 $kHz$ repetition rate. Designed by Cimel Electronique, it utilizes a shared transmitter/receiver telescope (diameter of 20 $cm$ and half field of view of 55 $\mu rad$) connected via *Why is the optical fibre so long?* an 10 $m$ optical fiber to its laser source and control system (laser divergence of 240 $mrad$). *Why use a PDA instead of a 532 nm photomultiplier? Wouldn't the signal-to-noise ratio be higher?* 
[revised manuscript text omitted]

What is the integration time for this profile (1 min)?

[Figure]

180    **Figure 2.** Data quality control of lidar profiles. (a) Rayleigh fit procedure was applied on 2,567 profiles (10 % of total) when clean conditions were met and AOD from photometer was available. The slope index temporal series (b) and histogram (c) are presented, along with the derived lidar constant ($C_L$) temporal series (d) and histogram (e). The AOD data used for the lidar constant calculation correspond to level 1.5 data of photometer B (#1243).

Figure 2 resumes the lidar data quality control performed for TRANSAMA data. From a total of 26,215 one-minute profiles generated during the campaign, a set of 2,567 Rayleigh fits (Fig. 2(a)) have been performed when clean conditions were met, i.e., no clouds and AOD at 532 nm below 0.1. Coincident photometer measurements are accounted within a 5-minute threshold. The reference zone $r_{ref}$ was defined between 5 and 7 km a.s.l. (above sea level), and the molecular coefficients were calculated using the tropical standard atmosphere. From the Rayleigh fits performed, slope indexes (Fig. 2b and 2c) and the lidar calibration constant (Fig. 2d and 2e) were calculated. This analysis confirmed the good performance of the lidar system during the campaign, with a slope index showing an overall low bias of 4% relative to the molecular attenuated backscatter profile and a 16% error margin despite the considerable noise induced by solar irradiance. However, frequent laser attenuation due to sea spray deposition on the lidar window prevented a consistent lidar constant for the dataset, with values varying a 30 % relative to the mean ($C_L = (4.3 \pm 1.4) \times 10^{18} \ photons/s \ m^3 sr$).

Therefore, a normalized relative backscatter (NRB) was defined instead (Lopatin et al., 2013).

$$NRB(r) = \frac{RCS(r)}{\int_{r_o}^{r_{max}} RCS(r\prime)dr\prime} \tag{3}$$

Where $r_o$ is the lower detection limit (Sect. 3.1.2), and $r_{max}$ is set at 10 km consistent with prior studies (Welton and Campbell, 2002; Lopatin et al., 2013, 2024). In particular the NRB profiles improve studies on atmospheric structure and data visualization, but do not have an impact on the later inversion procedure (Section 3.3) while the detection limits are not strongly affected.

**3.1.2 Detection limits**

The detection limits of the CE370 lidar primarily depend on signal-to-noise ratio (SNR) and the overlap function as discussed in previous studies (Popovici et al., 2018; Sanchez Barrero et al., 2024). The lower detection limit (blind zone) was defined at 350 m, where overlap function uncertainties exceed 20 %. The upper detection limit, determined by SNR < 1.5, varies between 10 and 18 km due to background noise from solar irradiance and the laser beam attenuation. Lidar profiles cannot be inverted if the SNR is 1.5.

Significant motion of the RV caused by rough seas, resulting in high variation in pitch or roll, introduces negligible uncertainties in the lidar profiling. For example, a 5-degree tilt in any direction causes altitude errors of 0.3% and horizontal shifts of 0.1% relative to the zenith. Furthermore, the system's combination of a wide laser divergence and a narrow telescope field of view minimizes misalignments due to platform motion. Additionally, the lidar's slightly offset viewing angle reduces direct reflections from clouds, enhancing measurement quality. This increases the recovery distance and accessibility to the lower layers, which contain the most sea salts.

[revised manuscript text omitted]

**3.3 Deriving aerosol properties profiles**

The study employs the Klett-Fernald inversion method (Klett, 1981, 1985; Fernald, 1984) to solve the underdetermined lidar equation (Eq. 1). By defining the aerosol extinction-to-backscatter ratio (lidar ratio, LR), the equation is reformulated as a Bernoulli differential equation(Weitkamp, 2005; Speidel and Vogelmann, 2023). The solution depends on boundary conditions defined by the position of an altitude reference zone:

- Backward solution: Reference zone in the far range, dominated by molecular contributions (same as Rayleigh Fit).
- It is an unstable solution from a mathematical point of view.
  Forward solution: Reference zone near the ground, where aerosol attenuation can be derived or approximated to 1.

In the case of the TRANSAMA campaign, only backward solution is considered, based on daily variations in the range detection limits, as discussed in Sanchez-Barrero et al. (2024). The effective LR (i.e., vertically constant) was iteratively varied to constrain the solution using column-integrated AOD from photometer measurements. As results, vertical profiles of aerosol BSC ($\beta_{aer}$ in m$^{-1}$ sr$^{-1}$) and EXT ($\alpha_{aer}$ in m$^{-1}$) are derived (Mortier et al., 2013; Popovici et al., 2018). Molecular BSC and EXT profiles were derived from GDAS (Global Data Assimilation System) data or a tropical standard atmosphere model. GDAS meteorological data is publicly available through NOAA Air resources laboratory (https://www.ready.noaa.gov/READYamet.php, last access: May 9 2025).

The approach using the coupling between a photometer and a lidar was used during INDOEX (doi:10.1029/2002JD002074).

**3.3.1 Uncertainties**

[revised manuscript text omitted]

---

## Author Comment (AC3)

**General answer to editor and referees' comments**

We would like to thank the two referees and editor for their insightful comments and constructive feedback, which are highly appreciated. Your thorough review has helped us refine the presentation of the manuscript and strengthen our conclusions.

We fully agree that Figures 3 and 4 contained too much information, making them difficult to interpretate, and we recognize that not all the details were directly relevant to this study. Therefore, substantial modifications have been made to improve both the structure of the paper and the readability of the results.

In particular, Section 3 has been revised to include only the methodology descriptions, while the instrumental assessments (data quality control under real conditions) have been moved to Section 4 (*Results and Discussion*). This restructuring also allows us to expand the discussion on current and future developments of both the lidar and photometer systems aboard vessels.

- 1. Introduction
- 2. Remote sensing instrumentation
  - 2.1. Single-wavelength lidar
  - 2.2. Photometers
  - 2.3. Ancillary measurements
- 3. Methodology
  - 3.1. Lidar data processing
    - 3.1.1. Data quality control and normalization
    - 3.1.2. Detection limits
    - 3.1.3. Atmospheric structure detection
  - 3.2. Photometer data processing
    - 3.2.1. Data quality screening analysis
  - 3.3. Deriving aerosol properties profiles
    - *3.3.1. Uncertainties*
- 4. Results and Discussion
  - 4.1. Local and synoptic scenario
  - 4.2. Instrumental Assessments
    - 4.2.1. Lidar data quality control
    - 4.2.2. Photometer data screening performance
  - 4.3. Aerosol variability during TRANSAMA
    - *4.3.1.* Atmospheric structure
    - 4.3.2. Transatlantic transport
- 5. Conclusions and Perspectives

Detailed responses to the individual comments are provided in the following pages. The editor (EC) and referees' (RC) comments are listed below in black and the authors' answers are listed in blue. The figures added within the responses are named as Figure R.

EDITOR COMMENTS .......... PAGE 2

ANONYMOUS REFEREE #1 ........ PAGE 3

REFEREE #2: CHAZETTE ....... PAGE 9

REFERENCES ......... PAGE 17

**Editor: Lionel Dopler**

I thank the authors of the paper "Aerosol variability over oceans using micro-pulse lidar and photometer: Insights from the TRANSAMA ship-based campaign" for their excellent work and valuable contribution. I fully agree with the reviewers that the manuscript convincingly demonstrates the synergy of lidar and photometer for the remote sensing of aerosols. In my opinion, it fits well within the scope of AMT and particularly this special issue. I also consider that the study and its survey are of high quality.

Nevertheless, I share the reviewers' concern regarding Section 3, which is excessively long. It could be substantially shortened, potentially integrated into another section, allowing more emphasis on the discussion of the benefits of combining active remote sensing (lidar) with passive sensing (photometer), which constitutes the main strength of the paper. Figures 3 and 4, in my view, are not easily interpretable. I suggest providing accompanying tables with reducing the content to only the most relevant information, or removing them entirely if Section 3 undergoes major restructuring. The issues with these figures have also been noted by the reviewers.

Addressing these points should be the priority in any potential revision requested after acceptance.

I wish the authors every success in the revision process and sincerely thank and congratulate them for their work.

We have carefully revised Section 3 to focus on the methodology, moving the instrumental assessments to Section 4 (Results and Discussion) and improving the readability of Figures 3 and 4. These changes also allow a stronger emphasis on the synergy between lidar and photometer measurements, as suggested.

**Anonymous Referee #1**

This is a short-term campaign (April–May 2023) that could prove useful for improving current techniques related to photometers installed on board mobile platforms. However, this task seems to have already been addressed in a previous article (Torres et al., 2025), which made use of three years of measurements from the same platform described in the present work, and which, in fact, includes the period covered by this article.

The main objective of this article, namely to improve our understanding of aerosol properties in a marine environment, appears rather ambitious given the very limited measurement period employed. Nevertheless, this article can provide valuable experience for the deployment of aerosol observations on oceanic platforms, which are essential to enhance our knowledge of their global-scale concentrations and of their possible interactions within cloud formation processes and other climatic processes.

Therefore, this is a paper that provides necessary and useful information for the scientific community, is well-written, and is supported by high-quality data, perfectly aligning with the scientific objective of this journal. I thus recommend its publication in AMT with some scientific and technical comments.

The formulation of the main objective in the current manuscript may be misleading for the reader. We agree that the main objective is to demonstrate the synergistic operation of lidar and photometer systems aboard vessels, with the potential to enhance our understanding of aerosol properties in marine environment. The revised manuscript takes into account these comments.

**General comments:**

I wonder whether the main objective of this study is to increase our knowledge of aerosols in marine regions. From my point of view, the importance of this article lies in the simultaneous measurement on board a vessel of a photometer (already presented in a previous article) and a lidar (a clear novelty introduced in this study). At first, I understood that this work was intended to lay the foundations for mobile measurements with both instruments, but after completing the reading I am not entirely sure what Section 3 of the article actually contributes. I do not understand Figures 3 and 4, and initially I thought that new quality-control level criteria adapted to ship-based measurements were being introduced. Later I realized that this was not the case. In short, I do not clearly understand the main objective of this article nor the information presented in Section 3.

What is clear, however, is the usefulness of the synergistic information presented in this article for performing real-time atmospheric monitoring during the vessel transects. A clear example is the information shown in Figure 6. Section 4 presents relevant and useful information for continuous monitoring, but it perhaps includes too many variables and explanations, which may cause the reader to lose track of the sequential objective of the paper.

We acknowledge the concerns regarding the clarity and contribution of Section 3. In response, Section 3 has been revised to focus on the methodology, while the instrumental assessments and data quality discussions have been moved to Section 4 (*Results and Discussion*), improving both clarity and readability. Figures 3 and 4 have been simplified to present only the most relevant information, ensuring they better support the discussion of measurement techniques. We are confident that these revisions make the objectives of the study clearer and more accessible to the reader.

**Scientific/Technical comments:**

1. Paragraph 45-56: In this paragraph, the authors describe the current efforts to fill the observational gaps that still exist in the study of marine aerosols. They list a series of interesting campaigns and technical efforts that form part of the state of the art. However, and somewhat surprisingly, they do not include the scientific article that reports on the three-year dataset obtained with the mobile device designed for the same vessel by the same authors. It is not until line 78 that this article is referenced. Please revise the state of art about marine photometric campaigns.

The state of art of marine photometric campaigns has been revised and updated in the new version of the article.

2. Line 60: Forest fires are not included here?

References to forest fires studies using mobile lidars are included. However, this part of the paragraph has been slightly modified including studies of transported aerosols (smoke and dust) over oceans and over land.

3. Line 63: Is there any example of mobile Raman lidars to be added here?

Yes, there are studies to be referenced in this part of the introduction. For instance, the multi-wavelength Raman Polly XT lidar has been deployed during campaigns aboard the Polastern research vessel (e.g., Bohlmann et al., 2018). Additionally, HSRL systems have been successfully operated aboard aircraft in numerous campaigns (e.g., Burton et al., 2013). See also Referee Chazette's comment (RC 6).

4. Line 116: Please include here the reference to Giles et al. (2019).

**Added.**

- 5. Line 122: This sentence seems incomplete. Maybe it has sense if "which" is replace by "from"? Improved.
  - 6. Line 132: The use of the connector "nevertheless" in this sentence appears somewhat confusing.

The sentence has been improved.

7. Line 132: Regarding the PLASMA measurements, although they are not included in the current manuscript, were they conducted successfully? Do the authors plan to report them in a future publication? In my view, if the availability of PLASMA measurements is mentioned in the text, it would be helpful to provide some additional details about them to inform the reader. However, this is merely a suggestion for clarity and completeness.

The PLASMA photometer installed during the TRANSAMA campaign was a prototype of the PLASMA-3 instrument, which was not yet fully prepared for the harsh open-sea conditions. Consequently, it was operated only intermittently for short test periods rather than continuous measurements. The initial evaluations of its performance remain at Level 0 (voltage signal), as further processing to higher levels is not possible without proper calibration. Nevertheless, these short tests provided useful technical insights, particularly regarding the instrument's Moon-tracking. For clarity, the corresponding part of the text has been revised to better reflect these details.

8. Line 141: If the objective of this paper is to lay the foundations for lidar measurements on board a mobile platform, I believe it is necessary to provide details on how frequently cleaning should be performed, as well as additional technical information that could be useful for other similar experiences.

More technical information for lidar measurements aboard the research vessel are discussed in Section 4.2.1 (Lidar data quality control).

9. Line 204: Is there any difference compared to the values found "on the ground"?

The upper detection limits observed during the campaign were higher than those typically measured over land (~15 km). However, a detailed comparison was not performed due to the substantially different environmental conditions between the measurements. For example, lidar measurements over land were often conducted in more polluted regions, where lower detection limits are expected (see Popovici et al., 2018).

10. Line 236: Is Eck et al. 2014 a correct reference to state here the definition of AERONET Level 1 algorithm? In this regard, I must admit that I find this paragraph generally confusing. Perhaps it stems from a misunderstanding on my part, but I understand here that AERONET Level 1 is based solely on triplet filtering according to a threshold. However, later on, in line 262, it is stated that there are additional requirements for a measurement to progress from Level 0 to Level 1, according to Giles et al. (2019). Could the authors please clarify this point?

To clarify, the description of the Level 1 protocols in this manuscript follows Giles et al. (2019), which describes the Version 3 AERONET data processing used in this study. This approach includes the triplet filtering as well as additional checks required for a measurement to go from Level 0 to Level 1. The reference to Eck et al. (2014) was only for general context, the procedures details and thresholds discussed in the text are based on Giles et al. (2019).

11. Line 240: The authors clarify that Level 2 is not yet available for mobile data. Consequently, the following sentence regarding the possible presence of aerosol or cloud layers suggests that, due to the absence of Level 2, such structures may result in erroneous measurements. Additionally, I wonder whether the term "uncertainty" is the most appropriate in this context, or if another term might better convey the intended meaning.

It is true, we changed uncertainty by erroneous measurement as suggested.

12. Line 250: Why is "Ve" used as the acronym for triplet measurements? The same for HPRe.

The variable Ve was originally used as an acronym because LO data are voltage (V) signal. However, since in literature it is mostly expressed as digital number (DN) (see for example: Giles et al., 2019), the triplet variability metric has been renamed  $RMSE_{DN}$  to avoid potential confusion. A similar adjustment was applied to the vessel attitude parameter, where the previous notation HPRe (Heading, Pitch, Roll error) is now presented as  $RMSE_{HPR}$ .

13. Sect. 3.2.1: At this point, this referee was expecting the definition of specific quality levels for ship-based photometric measurements. WS, HPRe, CC, and Ve are introduced, and Figures 3 and 4 are presented. I have many questions regarding this.

We thank the reviewer for the comment. The objective of the multifactor analysis is to highlight the need for a more robust evaluation of the effects of vessel motion on ship-based photometer data quality and to motivate the potential development of improved protocols. We agree that this was not sufficiently clear in the original manuscript. To address this, Section 3.2.1 has been moved to Section 4 (Results and Discussion), where the discussion has been expanded, and the content of Figures 3 and 4 has been simplified for clarity.

14. In line 259, it is stated that these figures illustrate the filtering process. However, it seems that they actually provide an insight of the real conditions under which the AERONET criterion filters the data, is that correct? Or is a new process being introduced here?

We agree with the referee that the figures provide insight into how AERONET criteria filter data under real measurement conditions, rather than introducing a new process. The original text in Section 3.2.1 was not sufficiently clear on this point. In the revised manuscript, special attention has been given to clarifying the presentation and discussion of these results.

15. How are the 37% and 40% values in line 261 obtained? Are they averages of all data classified as subset frequency data L1?

Any Level 0 data that pass to Level 1 necessarily satisfy the triplet variance criterion. Therefore, the counts of Level 1 data meeting this requirement are equal to the total number of Level 1 measurements. The 37% and 40% values correspond to the proportion of L0 measurements that successfully passed to L1.

16. I may not have fully understood Figures 3 and 4 (a) and (b), but shouldn't the sum of all bars for each variable equal the total number of L0 and L1 data points? For example, in Figure 3(a), the total L0 is 3786. The sum of counts by WS is 3786, which is as expected. However, for HPRe it is 3737, and for Ve it is 7572. Could the authors please clarify this point?

The referee is correct that, in general, the sum of all bars for each parameter (except cloud coverage, where a single profile can contain both mid- and high-altitude clouds) should equal the total number of L0 or L1 data points.

- For HPRe (now referred to as  $RMSE_{HPR}$ ), some data points were missing due to a sign error in the code, which has been corrected for the revised manuscript.
- For Ve (now  $RMSE_{DN}$ ), the sum is doubled because  $RMSE_{DN}$  is calculated for two different wavelengths (440 and 870 nm).

These corrections have been implemented in the revised figures and manuscript text.

17. Subset frequency data refers to the amount of data passing and specific screening?

Yes, "subset frequency data" refers to the percentage of data points meeting specific threshold conditions that pass a given screening level. For example, it indicates the fraction of measurements with WS

Figure R1: Time series of AOD(440) and EAE(440-870) overlaid with apparent wind speed.

23. References: The reference to Torres et al. (202) appears to be incomplete.

Corrected.

**Referee #2: Chazette**

Although the geophysical conclusions drawn from this offshore campaign confirm existing knowledge, this article is relevant to AMT. The campaign demonstrates the value of lidar/photometer synergy in the regular monitoring of sparsely covered ocean areas. Such a deployment could complement Earth observation from space, as with the EarthCARE mission. This potential should be discussed.

The potential of the instruments should be given more emphasis, and the associated uncertainties should be discussed in greater detail. In particular, it is difficult to take measurements in a clean environment where AODs are often below 0.1. Given the significant uncertainty surrounding Ångström exponents and LR values, conclusions must take this limitation into account.

Using a combination of lidar and photometer instruments on vessels provides important additional data to existing land-based networks and spaceborne observations. The frequency with which such observations could be made should be estimated, taking into account the additional resources available beyond the Marion Dufresne vessel.

In response to the suggestions:

**Potential of instruments and uncertainties:** We have expanded the discussion to emphasize the capabilities of both lidar and photometer systems, considering low-AOD marine conditions, and we now include a more detailed description of uncertainties associated with EAE and LR.

**Support to satellite missions:** We have added information in the introduction on how ship-based lidar/photometer deployments complement spaceborne observations, including potential CAL/VAL assessments with missions such as EarthCARE.

**Observation frequency and practical deployment:** We now discuss the potential frequency of such measurements, considering the operational constraints and resources available with the Marion Dufresne vessel and beyond.

**Scientific/Technical Comments**

1. Line 19: It is a long journey. For example, how does the top of the boundary layer differ between coastal areas and the open ocean, or between the Indian and Atlantic oceans?

This discussion was addressed in the Section 4.2 (Now 4.3.1).

2. Line 20: The AOD values presented are unusual?

No, the values are not unusual for clean maritime environment. An extended discussion is presented in Section 4.

3. Line 22-23: Cloud layers covering a large range of altitudes (up to 16 km) were observed in 53% of the lidar profiles with higher occurrence in low altitudes where the aerosol content was higher. What in the article demonstrates this?

It is true that we cannot infer that aerosol content is directly related with the occurrence of aerosol layers. This phrase has been modified. See below:

Cloud layers covering a large range of altitudes (up to 16 km) were observed in 53% of the lidar profiles, with a higher frequency at lower altitudes, where aerosol layers were more frequently detected.

4. In the abstract, it would be better to emphasize the importance of regular observations using synergy of instruments, given that there have been no new discoveries regarding how atmosphere works.

We agree with the referee comment and have slightly revised the abstract to emphasize the importance of observations using the synergy of instruments, highlighting their value for improving the characterization of aerosol properties over oceans.

5. Line 32: See also: Flamant, C., Trouillet, V., Chazette, P., and Pelon, J.: Wind speed dependence of atmospheric boundary layer optical properties and ocean surface reflectance as observed by airborne backscatter lidar, J. Geophys. Res., 103, 25137–25158, https://doi.org/10.1029/98JC02284, 1998.

**Added as reference.**

6. Line 63: I don't see why Raman lidar would be any more dependent than a Rayleigh -Mie lidar. It depends on the components used. For HSRL, there are greater thermal constraints, but we are making progress. Several HSRL and Raman lidars are already operating in aircraft.

We agree that the stability of a lidar system depends on the components used. It is true that developments toward more robust designs are progressing rapidly, and both HSRL and Raman systems are now successfully operated on aircraft during campaigns. However, most high-power, specialized lidar systems, are bulky systems, that still require regular maintenance and calibration, which makes unsupervised and continuous operation challenging. In this context, and following the advances in photometry toward establishing a network over oceans (Torres et al., 2025), we propose the use of micro-pulse lidar systems, which are compact, robust, do not require regular maintenance (change of flash-lamps, alignment, regular calibration etc.) and are adapted for autonomous operation. This section in the introduction has been slightly modified to clarify the statements.

7. Line 88: The primary aim of the study is to demonstrate the benefits of combining LiDAR and photometer technology on vessels.

Yes, we agree. The main objective of the study is to demonstrate the benefits of combining lidar and photometer measurements on vessels. The corresponding paragraph has been revised to improve clarity and better convey this aim.

8. Regarding the objective of article: There is insufficient data for a study of aerosols above the oceans. This is an example of a potential application.

**Same as response for RC 7.**

9. Paragraph Lines 101-103: This paragraph should be placed at the end of the introduction.

**Done**

10. Line 107: Why is the optical fiber so long?

The 10 m optical fiber is used to separate the LIDAR telescope (housed outdoors in a thermal enclosure) from the protected opto-electronics and PC parts inside. This configuration is especially advantageous at sites with limited space or restricted access to nearby power outlets and equipment rooms.

11. Line 108: Why use a PDA instead of a 532 nm photomultiplier? Wouldn't the signal-to-noise ratio be higher?

We agree with the referee that the SNR would be slightly better but the stability (both mechanical and electronical) of the APD is much better for an industrial system. The received signals are weak since this is a micro-pulse lidar. The APD counter is designed to detect small signals and offers good detection efficiency. It is temperature-stabilized, has low dark-count noise, and operates on a 5 V DC supply.

With a photomultiplier, a supply voltage of several hundred volts is required. It is also more sensitive to temperature and magnetic fields.

12. Section 2.2.: Provide an overview of the uncertainties surrounding AODs and EAEs.

More details on the uncertainties are included in Section 3.2.

13. Re-organize Section 2: The description of the instrumentation setup (Sect. 2.3) can go at the beginning of the section 2. Sections 2.1 (lidar) and 2.2 (photometers) can be maintain and add a Section 2.3 for Ancillary data.

We took the suggestions on the re-organization of Section 2.

14. Line 162: Separate the different members of the equation.

The description of the different members of the equation is provided in the text.

15. Line 166: eliminate EXT.

Erased.

16. Line 172: Note: Rayleigh fit procedure is very classic, even before 2018.

Noted.

17. Line 173: An apostrophe missing for r in the integration term.

Corrected.

18. Figure 2: Are there any aerosols or semi-transparent clouds above the reference area? The slight difference in the molecular gradient above 7 km could be due to thin clouds or a changing baseline.

Yes, as discussed later in Sect. 4.2 (Now Sect. 4.3.1), the presence of clouds may be slightly underestimated due to the thresholds applied during automatic detection. Thin cirrus clouds partially masked by noise were difficult to identify, which may have affected the Rayleigh fit and subsequent inversion. To improve inversion results, additional filtering was applied, including the exclusion of profiles with lidar ratios exceeding 100 sr or negative extinction coefficients within the first kilometers.

19. Figure 2 (a): What is the integration time for this profile (1 min)?

Yes, the integration time for each profile is 1 minute.

20. Line 203: Lidar profiles cannot be inverted if the SNR is 1.5.

We agree, therefore the detection limit is imposed at SNR=1.5.

21. Line 207: This increases the recovery distance and accessibility to the lower layers, which contain the most sea salts.

The suggested text has been added.

22. Section 3.1.3: It is not very clear. Are two different algorithms used to find atmospheric structures? However, despite these fairly standard algorithms, false detections and non-detections often occur. This depends heavily on the transitions between layers.

No, only one algorithm was used for cloud detection, which follows the guidelines of the STRAT and BASIC algorithms. The corresponding section has been revised for clarity.

23. Line 254: Is there a reference?

The idea behind this study is inspired by Giles et al. (2019), who presented a comprehensive evaluation of AERONET's screening protocols, showing failure rates at each level protocol and proposing new tests to enhance data quality control. In our work, we focus on the parameters most likely to affect the progression of ship-borne photometer measurements through the quality levels.

24. Section 3.2.1: Figure 4 presents the photometer results differently than in Torres et al. (2025), but what new information does it provide?

Torres et al. (2025) discussed the impact of vessel motion on data quality, considering a 5-minute threshold for a single-case retrieval. In contrast, the present study (Figures 3 and 4) examines the influence of vessel motion on AOD data level transitions over the entire campaign and explores correlations with additional parameters such as wind speed and cloud coverage. In the revised manuscript, the figures have been simplified and the discussion expanded to highlight the potential for further instrumental assessments and the possible evaluation of data quality protocols specifically adapted for ship-based photometers.

25. Line 295: It is an unstable solution from a mathematical point of view.

We agree, added to the text.

26. Section 3.3: The approach using coupling between a photometer and a lidar was used during INDOEX (doi:10.1029/2002JD002074)

We added the reference in the introduction.

27. Section 3.3.1: So, what are the uncertainties regarding extinction profiles and LR?

LR uncertainty was constrained by matching AOD uncertainties ( $\pm 0.01$ ) in the iterative solution. For low-AOD conditions (AOD

Figure R2: Comparison of HYSPLIT back-trajectory results using the standard model and the ensemble mode for the time interval T3 at  $\sim$ 03:00 on May 5,2023.

43. Figure 9: Are the lower points extrapolations within the range of overlap?

Yes, the lower points correspond to the lidar blind zone below 350 m, where overlap errors of approximately 20 % are calculated (see Sect. 3.1.2).

44. Section 4: What conclusions have been drawn, in particular following the CALIPSO/CALIOP mission.

Unfortunately, during the campaign, the Marion Dufresne RV did not coincide exactly with the CALIOP spaceborne lidar trajectory while traversing the South Atlantic Ocean. However, on May 5 we identified the closest CALIOP trajectory in latitude, which was separated by 10° in longitude (~800 km east) from the vessel during the time interval T3 (see Fig. R2). The corresponding quicklooks showed signatures of biomass burning aerosol layers at altitudes similar to those observed in our measurements (see Fig. R3, red boxes indicate the CALIOP profiles closest to the vessel trajectory), further supporting the aerosol types reported in our study.

**Figure R3:** CALIOP quicklooks of total attenuated Backscatter and aerosol subtype mask. Red boxes indicate the profiles closest to the vessel trajectory and white dashed line highlights the presence of aerosol layer classified as biomass burning aerosols.

45. Line 479: replace were with may be.

**Done.**

46. Line 480: add seem after LR values and replace findings with conclusions.

**Done.**

47. Paragraph Lines 488-492: That is the most important highlight of the article.

We agree with the reviewer. The main conclusion has been emphasized and is now presented as the first bullet point in the revised manuscript.

**References**

Bohlmann, S., Baars, H., Radenz, M., Engelmann, R., and Macke, A.: Ship-borne aerosol profiling with lidar over the Atlantic Ocean: from pure marine conditions to complex dust—smoke mixtures, Atmos. Chem. Phys., 18, 9661–9679, https://doi.org/10.5194/acp-18-9661-2018, 2018.

Burton, S. P., Ferrare, R. A., Vaughan, M. A., Omar, A. H., Rogers, R. R., Hostetler, C. A., and Hair, J. W.: Aerosol classification from airborne HSRL and comparisons with the CALIPSO vertical feature mask, Aerosols/Remote Sensing/Validation and Intercomparisons, https://doi.org/10.5194/amtd-6-1815-2013, 2013.

Giles, D. M., Sinyuk, A., Sorokin, M. G., Schafer, J. S., Smirnov, A., Slutsker, I., Eck, T. F., Holben, B. N., Lewis, J. R., Campbell, J. R., Welton, E. J., Korkin, S. V., and Lyapustin, A. I.: Advancements in the Aerosol Robotic Network (AERONET) Version 3 database – automated near-real-time quality control algorithm with improved cloud screening for Sun photometer aerosol optical depth (AOD) measurements, Atmos. Meas. Tech., 12, 169–209, https://doi.org/10.5194/amt-12-169-2019, 2019.

Popovici, I. E., Goloub, P., Podvin, T., Blarel, L., Loisil, R., Unga, F., Mortier, A., Deroo, C., Victori, S., Ducos, F., Torres, B., Delegove, C., Choël, M., Pujol-Söhne, N., and Pietras, C.: Description and applications of a mobile system performing on-road aerosol remote sensing and in situ measurements, Atmospheric Measurement Techniques, 11, 4671–4691, https://doi.org/10.5194/amt-11-4671-2018, 2018.

Sanchez-Barrero, M. F.: Development of an autonomous integrated mobile system combining lidar and photometer to monitor aerosol properties in near real time, Earth Sciences. Université de Lille, 2024. English. (NNT: 2024ULILR014). (tel-05010974)

Smirnov, A., Sayer, A. M., Holben, B. N., Hsu, N. C., Sakerin, S. M., Macke, A., Nelson, N. B., Courcoux, Y., Smyth, T. J., Croot, P., Quinn, P. K., Sciare, J., Gulev, S. K., Piketh, S., Losno, R., Kinne, S., and Radionov, V. F.: Effect of wind speed on aerosol optical depth over remote oceans, based on data from the Maritime Aerosol Network, Atmos. Meas. Tech., 5, 377–388, https://doi.org/10.5194/amt-5-377-2012, 2012.

Sun, K., Dai, G., Wu, S., Reitebuch, O., Baars, H., Liu, J., and Zhang, S.: Effect of wind speed on marine aerosol optical properties over remote oceans with use of spaceborne lidar observations, Atmos. Chem. Phys., 24, 4389–4409, https://doi.org/10.5194/acp-24-4389-2024, 2024.

Torres, B., Blarel, L., Goloub, P., Dubois, G., Sanchez-Barrero, M. F., Popovici, I. E., Maupin, F., Lind, E., Smirnov, A., Slutsker, I., Chimot, J., González, R., Sicard, M., Metzger, J. M., and Tulet, P.: Adaptation of the CIMEL-318T to shipborne use: 3 years of automated AERONET-compatible aerosol measurements on board the research vessel Marion Dufresne, Atmos. Meas. Tech., 2025.